# New carbohydrate binding domains identified by phage display based functional metagenomic screens of human gut microbiota

Akil Akhtar[1,3], Madhu Lata[1,2], Sonali Sunsunwal[1,4], Amit Yadav[1,2,4], Kajal LNU[1,2,4], Srikrishna Subramanian[1,2] & T. N. C. Ramya [1,2 ✉]

Uncultured microbes represent a huge untapped biological resource of novel genes and gene products. Although recent genomic and metagenomic sequencing efforts have led to the identification of numerous genes that are homologous to existing annotated genes, there remains, yet, an enormous pool of unannotated genes that do not find significant sequence homology to existing annotated genes. Functional metagenomics offers a way to identify and annotate novel gene products. Here, we use functional metagenomics to mine novel carbohydrate binding domains that might aid human gut commensals in adherence, gut colonization, and metabolism of complex carbohydrates. We report the construction and functional screening of a metagenomic phage display library from healthy human fecal samples against dietary, microbial and host polysaccharides/glycoconjugates. We identify several protein sequences that do not find a hit to any known protein domain but are predicted to contain carbohydrate binding module-like folds. We heterologously express, purify and biochemically characterize some of these protein domains and demonstrate their carbohydrate-binding function. Our study reveals several previously unannotated carbohydrate-binding domains, including a levan binding domain and four complex N-glycan binding domains that might be useful for the labeling, visualization, and isolation of these glycans.

[1] CSIR- Institute of Microbial Technology, Sector 39-A, Chandigarh 160036, India. [2] Academy of Scientific & Innovative Research (AcSIR), Ghaziabad, Uttar Pradesh 201002, India. [3] Present address: Emory Vaccine Center, Emory University, Atlanta, GA, USA. [4] These authors contributed equally: Sonali Sunsunwal, Amit Yadav, Kajal LNU. ✉email: ramya@imtech.res.in

Earth is home to microbial communities of ~4–6 × 10^30 cells[1] of more than one trillion species[2], the vast majority of which have not been cultured by standard laboratory techniques[3] or studied. Microbial communities represent a potential reservoir for mining novel enzymes and biomolecules[4]. Sequencing-based as well as functional metagenomics, the application of culture-independent methods, have served as a powerful means of understanding and exploiting the complexity of microbial communities for the discovery of new biomolecules[5]. Functional metagenomics, which involves the construction of metagenomic DNA libraries (using suitable vectors such as cosmids, fosmids[6] or phages[7]) and their screening for a desired phenotype[8], enables the identification of genes that code for gene products with desired functions without any prior information about their properties based on the sequence similarity in public databases. Therefore, this strategy along with providing novel genes/proteins for biotechnological applications[8] also aids in the functional assignment of proteins designated as hypothetical proteins in databases[9].

The human gut is a glycan-rich landscape with enormous glycan structural complexity[10] arising from the huge diversity of monosaccharides and glycosidic linkages in the endogenous glycans of mucin[11,12] in the host mucus[13] as well as in dietary plant polysaccharides such as starch, hemicellulose, and pectin[10]. In contrast to human genomes, which encode only 97 glycoside hydrolases, with only 17 breaking down a small subset of glycosidic linkages present in carbohydrate nutrients like sucrose, lactose and starch, a mini-microbiome built with 177 reference genomes of the human gut microbiota was found to possess 15,882 different carbohydrate-active enzyme (CAZyme) genes responsible for glycan metabolism[14]. It is evident that the metabolism of complex carbohydrate nutrients in the gut is undertaken by the CAZymes of the roughly trillion microbes of ~160 species that make up the human gut microbiota[15–17]. CAZymes frequently have a modular architecture, with one or more catalytic domain(s) in tandem with ancillary non-catalytic modules or carbohydrate binding modules (CBMs)[18] that bring the catalytic module into proximity of inaccessible substrates and thus increase the effective substrate concentration and enzyme efficiency[19]. Besides CBMs, other proteins such as bacterial lectins also bind to carbohydrate molecules in the gut[20]. Thus, the human gut microbial metagenome is a huge treasure trove of genes encoding carbohydrate binding proteins that can be exploited for a variety of applications[21] like blood typing[22], biospecific affinity purification[23], and targeting applications for anti-tumor and anti-viral therapy[24], and that may also further our understanding of host-commensal interactions.

We report here the validation of a biopanning procedure for the identification of carbohydrate binding proteins using a fucose-binding phage, SrNaFLD-T7, the construction of a metagenomic phage display library, and biopanning against a panel of different glycans and glycoconjugates, culminating in the identification of previously unannotated carbohydrate binding domains from the metagenome of healthy human gut microbiota.

## Results

**Strategy of study**. The overall strategy of our functional metagenomics approach, as outlined in Fig. 1a, was to use the direct physical link between the genotype and the phenotype provided by the phage display system to identify microbial carbohydrate binding protein domains present in the human gut milieu. We considered the following in selection of a phage display vector— (1) the median size of bacterial proteins is 267 amino acids[25], (2) CBMs are typically about 30 to 200 amino acids long[23], (3) T7 phage display systems have less sequence bias as compared to M13 and can be used to display proteins of about 1200 amino acids[26,27], (4) individual protein-carbohydrate interactions are typically weak and require multivalency for optimal binding strength. We chose to construct the phage display library using the commercially available phage vector T7Select 10-3b to maximize on the display size (~1200 amino acid long peptides) and simultaneously have a sufficient copy number of peptides (5–15 per phage) in order to enable avidity of binding and thereby efficient screening. We constructed a metagenomic phage display library with DNA pooled from fecal samples of healthy Indian subjects from four geographical locations in Northern, Northwestern and Central India. We did this in order to increase the diversity of genes encoding for microbial carbohydrate binding domains in the metagenomic library. We subjected the metagenomic library to biopanning against various carbohydrates/glycoconjugates (Supplementary Data 1, Table S1) in order to enrich for and identify new carbohydrate binding protein domains. For biopanning, we selected mucin (host glycoprotein) and bacterial peptidoglycans in order to identify microbial glycan binding proteins that enable host intestinal colonization and intra/interspecies interactions in the gut milieu. We also selected plant carbohydrates such as starch, inulin, and β-D-glucan that are present in food grains and edible tubers, etc. in order to identify microbial carbohydrate binding protein domains that enable utilization of dietary plant polysaccharides.

**Validation of biopanning protocol using an L-fucose-binding T7 phage (SrNaFLD-T7Select 10-3b phage)**. Considering that biopanning of phage display libraries is laborious and prone to the identification of false-positives[28], we optimized our biopanning protocol for protein-carbohydrate interactions with a recombinant L-fucose binding T7 phage. We cloned the Na and fucose-binding F-type lectin domains (NaFLD) of the previously characterized α-L-fucosidase protein from *Streptosporangium roseum* (*S. roseum* DSM 43021)[29] into the T7Select 10-3b phage to obtain the recombinant L-fucose binding SrNaFLD-T7Select 10-3b phage (Fig. S1). We biopanned phage mixtures of SrNaFLD-T7Select 10-3b phage and non-recombinant T7Select 10-3b phage in various ratios against mucin, eluted bound phages with L-fucose, and fed them into subsequent biopanning cycles (Fig. S1, Supplementary Data 1, Table S2). We followed the phage titer after each cycle of biopanning and screening randomly selected phage plaques for the presence of a ~900 bp PCR amplicon band corresponding to SrNaFLD. We thus ascertained that we could effectively enrich a fucose-binding recombinant phage clone present at a copy number of just 100 (amidst a total of ~10^10 phages) by our phage display selection. Our results indicated that six or more cycles of phage display selection would be required for the effective enrichment to ~10% of low abundance genes in a phage display library coding for glycan binding proteins (Fig. S1, Supplementary Data 1, Table S2). We also found that L-fucose (100 mM), the natural ligand of SrNaFLD, was better than 1% SDS, the commonly used elution agent for disrupting protein-ligand interactions, at enriching for L-fucose-binding phages (Fig. S1).

**Construction of human fecal metagenomic phage display library and biopanning against carbohydrates/glycoconjugates**. We successfully ligated 500–3000 bp long human fecal metagenomic DNA fragments to T7Select 10-3b phage vector arms (Fig. S2), to obtain a human fecal metagenomic phage display library of 1.2 × 10^7 phages, of which five sixths (1 × 10^7 phages) were recombinants (Fig. 1b). Considering an average insert size of 1 kbp and an average bacterial genome size of ~5 Mbp[30], we estimate that the library accommodates around 10 Gbp of

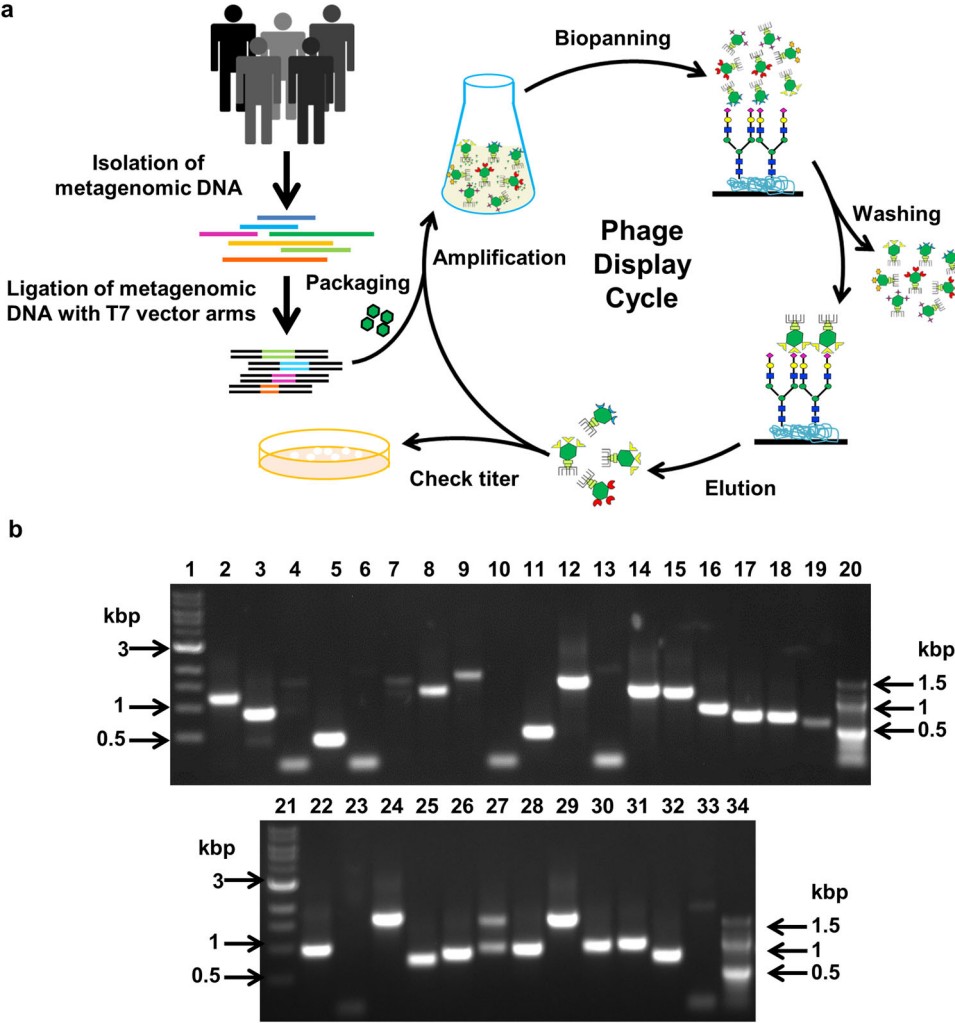

**Fig. 1 Metagenomic phage display library. a** Schematic representation of metagenomic phage display library preparation and biopanning to screen ligand binding recombinant phages. **b** Agarose gel electrophoresis of PCR amplicons indicating the presence or absence of any recombinant insert within randomly selected phages of the metagenomic phage display library. Results of a single screening experiment. Lanes 1 and 21: 1 kbp DNA marker (NEB). Lanes 20 and 34: 100 bp DNA marker (NEB). Lanes 2–19 and 22–33: PCR amplicons of the phages.

metagenomic DNA with diversity equivalent to ~2000 bacterial genomes. Assuming one sixth of the genes present in the inserts of this library are in the correct coding frame, the functional diversity of the library is anticipated to be ~1.66 Gb, which is equivalent to ~333 bacterial genomes.

We performed eight cycles of phage display selection against various carbohydrates/glycoconjugates with this metagenomic phage display library using a multiplicity of screening of 3000 and using various component monosaccharides or the polysaccharide as the eluting agent (Supplementary Data 1, Table S1), and screened randomly picked phage plaques by PCR for the presence of recombinant inserts. Typically, the phage titres in the eluates obtained after these eight cycles of phage display selection were high (Supplementary Data 1, Table S3), and most randomly picked phage clones had T7 PCR amplicons >144 bp, indicating enrichment. A high degree of selection was also evident from the high frequency of occurrence of PCR amplicons of the same size among randomly picked phage clones for a given selection condition and sometimes, even for phage clones from different elution conditions (e.g. Fig. 2a–f). We confirmed this by sequencing a few amplicons of the same size and verifying that they had the same sequence. Sequences of all recombinant phage clones identified are listed in Supplementary Data 1, Table S4.

**Identification of glycan binding proteins by phage display selection against host-derived glycoconjugates (mucin).** Following eight cycles of biopanning against mucin from porcine stomach (Fig. 2a–f), we determined five unique insert sequences — MG1 from phage clones eluted with D-galactose, GalNAc, and sialic acid, MN3 from phage clones eluted with L-fucose and D-mannose, MU1 and MU3 from phage clones eluted with GlcNAc, and MF12 from phage clones eluted with L-fucose (Supplementary Data 1, Table S4).

Sequence analysis indicated that MG1 was similar to proteins from *Prevotellamassilia timonensis*, *Prevotella* sp. CAG:5226, and *Muribaculaceae bacterium*, all of which are annotated by Pfam to have a glycosyl hydrolase family 65, N-terminal domain (Supplementary Data 1, Table S5). All three organisms are gut commensals, and the presence of a similar sequence in these glycosyl hydrolase family 65, N-terminal domain containing proteins adds support for a glycan-binding function in the gut milieu for MG1. Sequence searches with MN3 and MU1 returned uncharacterized proteins and a lipocalin-like domain-containing protein, all from *Prevotella* species, and the search with MU3 returned MU1 as a hit, in addition to these. No hits were obtained for the sequence MF12 with the expected frame of translation (listed as MF12F1 in Supplementary Data 1, Table S4). However,

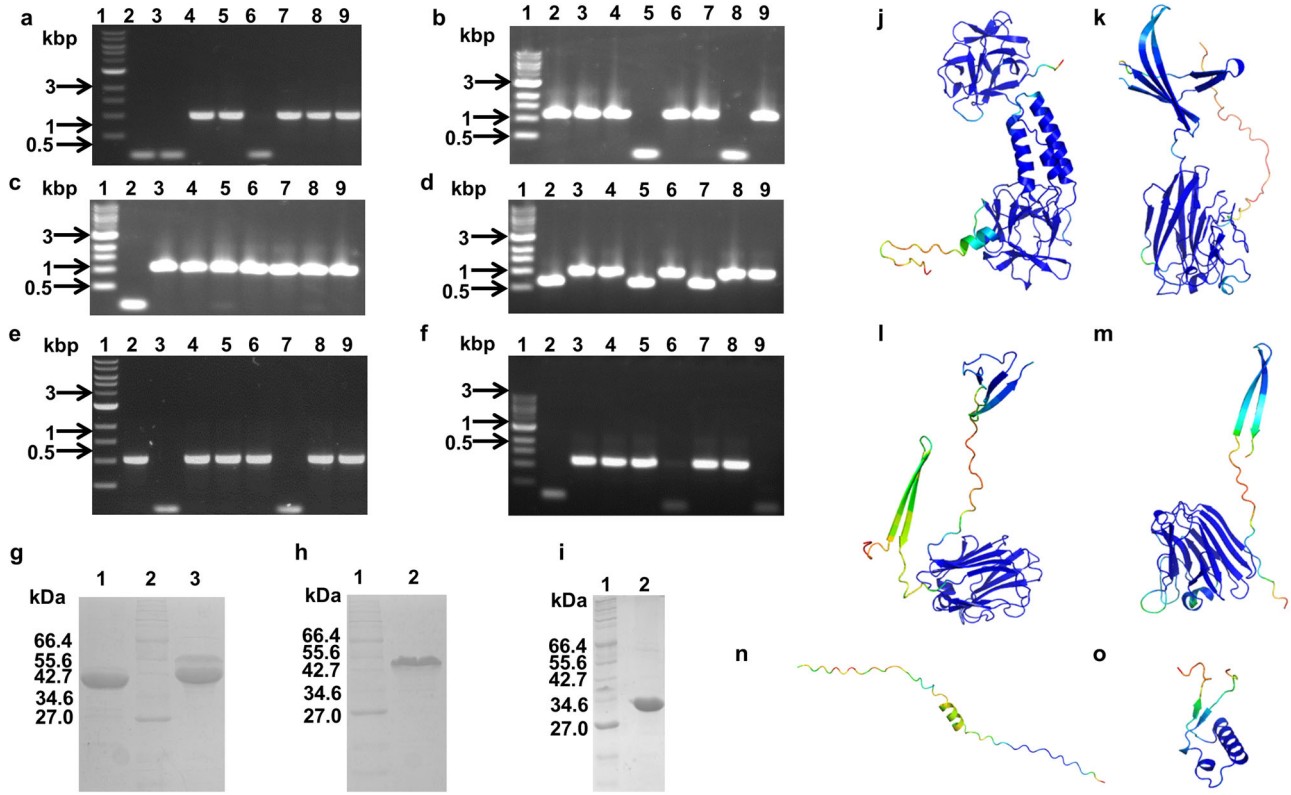

**Fig. 2 Identification of mucin binding protein domains. a–f** Agarose gel electrophoresis of PCR amplicons indicating the presence or absence of any recombinant insert following biopanning of the metagenomic phage display library against porcine stomach type II mucin. Results of a single screening experiment. **a** Lane 1: 1 kbp DNA marker. Lanes 2–9: Phages eluted with L-fucose, **b** Lane 1: 1 kbp DNA marker. Lanes 2–9: Phages eluted with D-galactose, **c** Lane 1: 1 kbp DNA marker. Lanes 2–9: Phages eluted with GalNAc, **d** Lane 1: 1 kbp DNA marker. Lanes 2–9: Phages eluted with GlcNAc, **e** Lane 1: 1 kbp DNA marker. Lanes 2–9: Phages eluted with D-mannose, **f** Lane 1: 1 kbp DNA marker. Lanes 2–9: Phages eluted with sialic acid (Neu5Ac), **g** Coomassie blue stained SDS-PAGE showing purified MG1 and MN3 proteins in the imidazole eluates following Ni-NTA metal ion affinity chromatography; Lane 1: Purified MG1 protein; Lane 2: Protein marker from NEB (2–212 kDa); Lane 3: Purified MN3 protein, **h** Coomassie blue stained SDS-PAGE showing purified MU1 protein in the imidazole eluates following Ni-NTA metal ion affinity chromatography; Lane 1: Protein marker from NEB (2–212 kDa); Lane 2: Purified MU1 protein, **i** Coomassie blue stained SDS-PAGE showing purified MU3 protein in the imidazole eluates following Ni-NTA metal ion affinity chromatography; Lane 1: Protein marker from NEB (2–212 kDa); Lane 2: Purified MU3 protein. **j–n** AlphaFold2 predicted structures of MG1 (**j**), MN3 (**k**), MU1 (**l**), MU3 (**m**), MF12 (**n**), and MF12F2 (**o**).

the amino acid sequence from translation frame 3 of the ORF (listed as MF12F2 in Supplementary Data 1, Table S4) returned transposase proteins as top hits (Supplementary Data 1, Table S5). We could not identify any conserved domain in any of these amino acid sequences—MG1, MN3, MU1, MU3, MF12F1, and MF12F2—by searching against Pfam[31,32] or the dbCAN2 database[33], indicating the absence of any annotated CAZyme, CBM or other protein domain (Supplementary Data 1, Tables S6, S7 and S8). An all against all protein sequence search indicated again a strong local sequence similarity between MU1 and MU3 (sequence identity of 39.2%) (Supplementary Data 1, Table S9).

We used the AlphaFold2 artificial intelligence software[34] to predict structural models of all these proteins. The MG1 structural model contains two β-trefoil-like domains (similar to ricin-type β-trefoil lectin domain) connected by a domain comprising three α-helices (Fig. 2j). MN3, MU1, and MU3 each contain a complete β-sandwich-like domain, similar to many CBMs and lectin domains, and partial β-strand containing domain(s) on the N- and/or C-terminus (Fig. 2k–m). A structural model could not be predicted for MF12F1 with reasonable confidence (Fig. 2n). MF12F2 was relatively more structured (Fig. 2o).

We successfully cloned MG1, MN3, MU1 and MU3 in pET-28a(+), and expressed and purified them by Ni-NTA affinity

chromatography from *Escherichia coli* BL21(DE3) cells (Fig. 2g–i, Fig, S3). Size exclusion chromatography indicated that MGI, MU1 and MU3 existed primarily as monomers (migrating as ~39 kDa, ~39 kDa and ~36 kDa species), and MN3 existed in the monomeric and dimeric forms (migrating as ~77 kDa and ~33 kDa species). Western blot analysis confirmed the presence of the C-terminal hexahistidine tag in MG1 and MN3, and of the N-terminal hexahistidine tag in all four proteins (Fig. S3). Intact mass analysis by ESI-MS yielded molecular masses expected upon N-terminal methionine cleavage, and CD spectroscopy indicated the presence of secondary structure (Fig. S3).

We profiled the glycan binding specificity of MG1, MN3, MU1, and MU3 by glycan micro-array experiments on the CFG glycan array version 5.3 (comprising 600 natural and synthetic glycans) (Supplementary Data 1, Table S10) followed by Glycopattern[35] (https://glycopattern.emory.edu) and MotifFinder[36,37] analysis. We found that MG1, MN3, and MU1 mostly bound to non-sialylated, terminal N-acetyllactosamine (LacNAc; Galβ1-4GlcNAc) moieties of complex bi-antennary and multi-antennary N-glycans (with or without α−1,6 core fucosylation) (Fig. 3, Table 1, Fig. S4, Supplementary Data 1, Table S10). MG1 preferred long N-glycan structures with multiple N-acetyllactosamine repeats ("i" antigen), terminating in N-acetyllactosamine or N-acetylneolactosamine (Galβ1-3GlcNAc) or GlcNAc (incomplete "i" antigen), and/or H

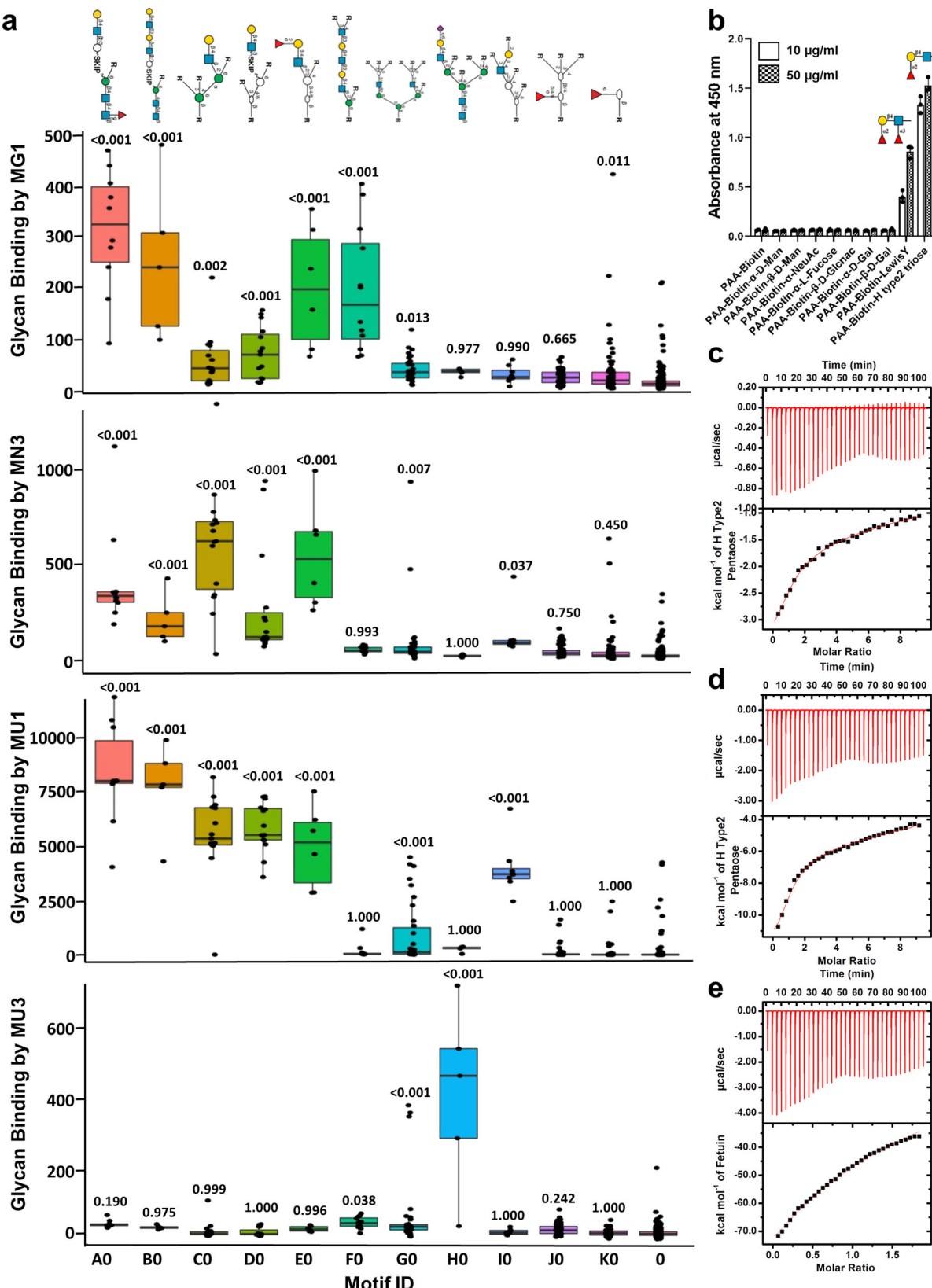

antigen (Fucα1-2 Gal) motifs. MG1 did not bind to long sialylated N-glycan structures or to short N-glycan structures. MN3 and MU1 bound to galactose terminating type-2 (Galβ1-4GlcNAc) glycan structures with a single N-acetyllactosamine unit and with multiple lactosamine units, respectively, and to glycans with H antigen type-2, Lewis^x and Lewis^y structures. MN3 and MU1 did not bind to N-glycans terminating in a neolactosamine unit, sialic acid or GlcNAc (incomplete "i" antigen), or to N-glycans containing A (GalNAcα1-3(Fucα1-2)Gal) or B (Galα1-3(Fucα1-2)Gal) antigen motifs. MU3 showed strict recognition of α2-6-linked sialic acids on short (single N-acetyllactosamine unit), multi-antennary, complex N-glycans with sialic acids on all

**Fig. 3 Glycan binding by MG1, MN3, MU1, and MU3. a** Bar plot indicating glycan binding intensities of motifs recognized by MG1, MN3, MU1, and MU3 (output of MotifFinder[36,37]). The motifs are represented in standard glycan symbol nomenclature[83] above the plot. The motif ID "0" refers to non-binders. Results of MotifFinder analysis using glycan array datasets of single runs of MG1 at 5, 50, and 400 μg/ml, and MN3, MU1, and MU3 at 5 and 50 μg/ml. Models were built manually for each protein using a common custom motif list obtained by combined automated model building using all glycan array datasets. *P*-values are calculated by the MotifFinder software and refer to difference from non-binders with multiple testing corrections (Dunnet's Test). **b** Binding of MN3 to Lewis$^y$ and H antigen type-2 triose as assessed by enzyme linked lectin assays. Results of a single experiment. Means are plotted. Error bars represent standard deviations of three technical replicates. Source data in Supplementary Data 3. **c** Binding of MN3 to H antigen type-2 pentaose as assessed by isothermal calorimetry. Results of a single run. **d** Binding of MU1 to H antigen type-2 pentaose as assessed by isothermal calorimetry. Results of a single run. **e** Binding of MU3 to Fetuin as assessed by isothermal calorimetry. Results of a single run.

**Table 1 Ligand binding profile of clones (MG1, MN3, MU1, and MU3) identified by biopanning against mucin**

| Ligands tested in binding assays | Ligands bound by | | | |
| --- | --- | --- | --- | --- |
| | MG1 | MN3 | MU1 | MU3 |
| Glycan Array (CFG glycan array version 5.3 comprising 600 natural and synthetic glycans)[a] | Asialo, long, complex N-glycans; terminal α1,2-Fuc | Asialo, short, complex N-glycans; terminal LacNAc; H, Le$^x$, Le$^y$ antigens | Asialo, long, complex N-glycans; terminal LacNAc; H, Le$^x$, Le$^y$ antigens | Sialo, short, complex N-glycans; terminal α2,6-Sia |
| ITC (LNnT, LNT, H type-2 pentaose, Fetuin, Asialofetuin, H type2 heptaose, Le$^a$, difucohexaose, Le$^b$, Le$^x$, H type-2 triose$^b$, LNnO$^c$, LNnH$^c$, A type-2 tetraose$^c$) | None | H type-2 pentaose | H type-2 pentaose | Fetuin |
| ELLA (PAA-Biotin and PAA-Biotin-linked α-D-Mannose, β-D-Mannose, α-Neu5Ac, α-L-Fucose, β-D-GlcNAc, α-D-Galactose, β-D-Galactose, Le$^y$, H type-2 triose) | Not tested | PAA-Biotin-Le$^y$, PAA-Biotin-H type-2 triose | Not tested | Not tested |

*ITC* Isothermal Calorimetry, *ELLA* Enzyme Linked Lectin Assay, *LNnO* Lacto-N-neooctaose, *LNnH* Lacto-N-neohexaose, *LNnT* Lacto-N-neotetraose, *LNT* Lacto-N-triaose, *A type-2 tetraose* A antigen type-2 tetraose, *H type-2 pentaose* H antigen type-2 pentaose-β-N-Acetyl Propargyl, *H type-2 heptaose* H antigen type-2 heptaose azido ethyl, *Le$^a$* Lewis A tetraose, *Difucohexaose* Lacto-N-difucohexaose II, *Le$^b$* Lewis B tetraose, *Le$^x$* Lewis X tetraose.
[a]Glycan array ligands represented are common motifs of glycans showing high binding intensity in the assay.
[b]Tested only with MN3.
[c]Tested only with MG1.

antennae, discriminating against multi-antennary N-glycans with α2-3 terminating sialic acids or asialylated antennae (Fig. 3, Table 1, Fig. S4, Supplementary Data 1, Table S10).

We detected binding of MN3 to H antigen type-2 pentaose by isothermal calorimetry (K$_D$ values of 36.76 μM and 2.9 mM; sequential two-sites model), and to H antigen type-2 triose and Le$^y$ hexaose (in the micromolar range) by enzyme-linked lectin assays (Fig. 3, Table 1, Fig. S5). We also detected binding of MU1 to H antigen type-2 pentaose (K$_D$ values of 20.83 μM and 6.25 mM; sequential two-sites model) and of MU3 to sialylated fetuin (K$_D$ of 239 μM) (Fig. 3, Table 1, Fig. S5). No binding was detected to any of the other glycans tested (Table 1, Fig. S5).

**Identification of amylose, amylopectin, and starch binding proteins.** We successfully identified the clones, Am-Glc2 (eluted from amylose with D-glucose), Am-Am1 (eluted from amylose with amylose), Ap-Ap2 and Ap-Ap5 (eluted from amylopectin with amylopectin), St-Glc1, St-Glc2, St-Glc5, and St-Glc8 (eluted from starch with D-glucose), and St-St4 and St-St5 (eluted from starch with starch) (Fig. 4a, b, Fig. S6, Supplementary Data 1, Table S4); other clones could not be successfully amplified. An all against all protein sequence search indicated >90% sequence identity amongst Ap-Ap2, Ap-Ap5 and St-Glc2, and amongst Am-Glc2, St-Glc1, and St-Glc5, with the clone Am-Glc2 (197 amino acids long, amylose binder eluted with glucose) mapping almost completely within St-Glc1 (370 amino acids long, starch binder eluted with glucose) (Supplementary Data 1, Tables S4 and S9).

Sequence searches indicated similarity of Am-Glc2, Am-Am1, Ap-Ap2, Ap-Ap5, St-Glc1, St-Glc2, St-Glc5, and St-St5 to proteins from *Prevotella* species and *Marseilla massiliensis*

containing starch binding module 26 (and α-amylase catalytic domain, in some cases) (Supplementary Data 1, Table S5). St-Glc8 and St-St4 showed similarity to outer membrane SusF_SusE domain containing proteins from *Prevotella* species (Supplementary Data 1, Table S5). The clones Am-Am1, Ap-Ap2, Ap-Ap5, St-Glc1, St-Glc2, and St-St5, and the clone, St-St4 also showed the presence of the Starch-binding module 26, and Outer membrane protein SusF_SusE domains, respectively, in searches against Pfam A, and the clones, Am-Glc2, St-Glc1, and St-St4 found hits in Pfam B (Supplementary Data 1, Tables S7 and S8). Searches against the dbCAN2 database indicated the presence of CBM26 in Am-Am1, Ap-Ap2, Ap-Ap5, St-Glc1, St-Glc2 and St-St5 (Supplementary Data 1, Table S6). AlphaFold2 predicted structural models showed the presence of β-sandwich domains, similar to CBMs and lectin domains (Fig. 4c–g, k, Fig. S6).

Recombinant St-Glc1 and St-Glc5v2 (sequenced clone differing from St-Glc5 in residues 61 (A instead of T) and 94 (S instead of C); Supplementary Data 1, Table S4) expressed and purified from *E. coli* BL21(DE3) cells (Fig. 4h, l) with molecular masses, 43.959 kDa and 25.037 kDa, respectively (as expected with N-terminal Methionine removal) (Fig. S7), demonstrated binding to starch and amylose (in the submicromolar range) but not to amylopectin (Fig. 4i, m, Table 2). Binding of St-Glc5v2 to immobilized starch and amylose decreased upon pre-incubation of protein with starch and amylose but not amylopectin or D-glucose whereas the effect was not as evident in St-Glc1 (Fig. 4j, n) and heat denaturation of protein resulted in loss of binding (Fig. 4j, n). These results further established that the method used by us was a reliable approach to screen the carbohydrate binding proteins/domains.

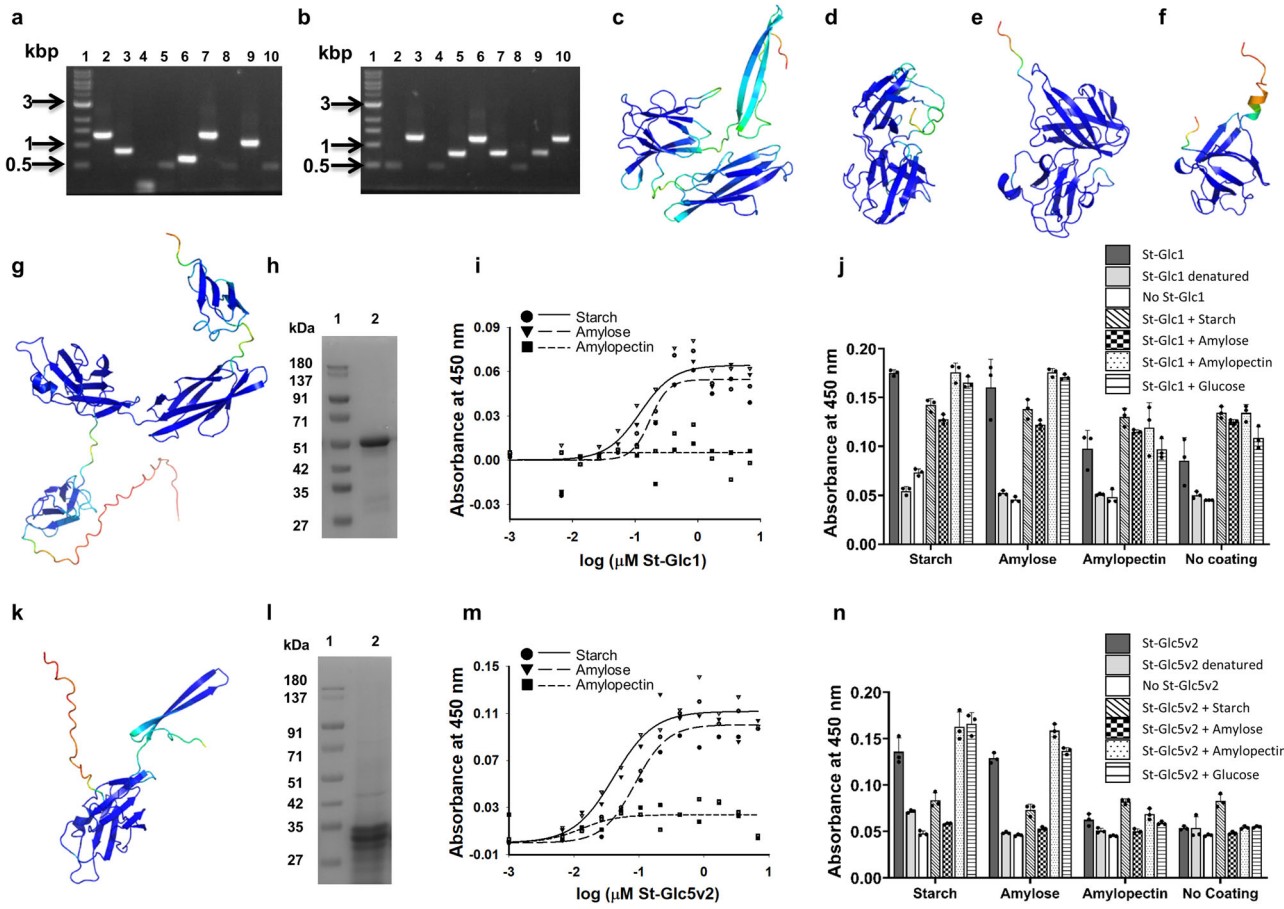

**Fig. 4 Identification of starch binding protein domains. a, b** Agarose gel electrophoresis of PCR amplicons indicating the presence or absence of any recombinant insert following biopanning of the metagenomic phage display library against starch. **a** Biopanning against starch and elution with D-glucose. Results of a single screening experiment. Lane 1: 1 kbp DNA marker. Lanes 2–9: PCR amplicons of clones St-Glc1 to St-Glc9. **b** Biopanning against starch and elution with starch. Results of a single screening experiment. Lane 1: 1 kbp DNA marker. Lanes 2–9: PCR amplicons of clones St-St1 to St-St9. **c–f** AlphaFold2 predicted structural models of St-Glc2 (**c**), St-Glc8 (**d**), St-St4 (**e**), and St-St5 (**f**). **g** AlphaFold2 predicted structural model of St-Glc1. **h** Coomassie blue stained SDS-PAGE showing purified St-Glc1 protein following Ni-NTA metal ion affinity chromatography; Lane 1: Protein molecular weight marker; Lane 2: Purified recombinant St-Glc1 protein. **i** Concentration dependent binding of St-Glc1 to immobilized starch, amylose, and amylopectin. Individual data points plotted after background subtraction along with regression curves. Results of a single binding experiment with two technical replicates. Source data in Supplementary Data 3. **j** Binding of St-Glc1 to wells containing immobilized starch, amylose, amylopectin, or buffer (blank) with or without pre-incubation with starch, amylose, amylopectin, or D-glucose. Results of a single binding experiment. Means are plotted as bars and individual data points included. Error bars represent standard deviations of three technical replicates. Source data in Supplementary Data 3. **k** AlphaFold2 predicted structural model of St-Glc5v2. **l** Coomassie blue stained SDS-PAGE showing purified St-Glc5v2 protein following Ni-NTA metal ion affinity chromatography; Lane 1: Protein molecular weight marker. Lane 2: Purified recombinant St-Glc5v2 protein. **m** Concentration dependent binding of St-Glc5v2 to immobilized starch, amylose, and amylopectin. Individual data points plotted after background subtraction along with regression curves. Results of a single binding experiment of two technical replicates. Source data in Supplementary Data 3. **n** Binding of St-Glc5v2 to wells containing immobilized starch, amylose, amylopectin, or buffer (blank) with or without pre-incubation with starch, amylose, amylopectin, or D-glucose. Results of a single binding experiment. Means are plotted as bars and individual data points included. Error bars represent standard deviations of three technical replicates. Source data in Supplementary Data 3.

**Identification of a novel levan binding protein domain.** We successfully amplified and identified the clones Lev-Lev5 (eluted from levan using levan) and Lev-Fru3 (eluted from levan with D-fructose) (Fig. 5a) and found them to be identical in sequence, indicating a high level of enrichment. Lev-Lev5 shared highest sequence similarity with uncharacterized proteins from *Prevotella* species that contained DUF4960, DUF5018, and DUF5006 (Supplementary Data 1, Table S5). A search against the dbCAN2 database did not return any hit and a search against Pfam returned the DUF4960 protein domain, confirming the absence of any annotated CAZy, CBM or other protein domain with characterized function (Supplementary Data 1, Tables S6, S7, S8). An all against all protein sequence search of all the sequences found

in this study did not reveal any similarity of Lev-Lev5 with any other clone (Supplementary Data 1, Table S9). The AlphaFold2 predicted structural model indicates the presence of a Rossmanoid Flavodoxin-like fold (Fig. 5b), and the multiple sequence alignment used by AlphaFold2 from PDB70 includes hypothetical glycoside hydrolases.

Recombinant Lev-Lev5 expressed and purified from *E. coli* BL21(DE3) cells (Fig. 5c, d) demonstrated appreciable binding to levan (from chicory) and less but significant binding to inulin (from chicory) at higher protein concentrations but not to β-D-glucan (from barley) or to other carbohydrates such as xylan, amylopectin, pectin, arabinogalactan, dextran, laminarin, amylose or starch; the apparent affinity for immobilized levan was

**Table 2 Ligand binding profile of clones identified by biopanning against various plant and bacterial polysaccharides**

| Clone[a] | Biopanning ligand / eluant | Whether purified protein tested in binding assays | Ligands tested in binding assay | Ligands bound by protein |
|---|---|---|---|---|
| St-Glc1 | Starch / D-glucose | Yes | Starch, amylose, amylopectin (ELLA) | Starch, amylose |
| St-Glc5v2 | Starch / D-glucose | Yes | Starch, amylose, amylopectin (ELLA) | Starch, amylose |
| Lev-Lev5 | Levan / D-fructose | Yes | Levan, inulin, xylan, amylopectin, pectin, arabinogalactan, glucan, dextran, laminarin, amylose, starch (ELLA) | Levan, inulin[b] |
| BDG-Glc5 | β-D-Glucan / D-glucose | No[c] | None | None |
| BDG-BDG6 | β-D-Glucan / β-D-Glucan | No[c] | None | None |
| AG-Gal2 | Arabinogalactan / D-galactose | No[c] | None | None |
| StapPG-GlcNAc6 | *Staphylococcus aureus* peptidoglycan / GlcNAc | Yes | *Staphylococcus aureus*, *Methanobacterium* sp., and *Saccharomyces cerevisiae* peptidoglycans (ELLA) | None |
| PBGal-Gal8 | PAA-β-D-Gal / D-galactose | Yes | PAA-biotin, biotin-PAA-β-D-galactose | None |
| PBGlcNAc-GlcNAc17F1 | PAA-β-D-GlcNAc / GlcNAc | No[d] | None | None |
| PBGlcNAc-GlcNAc4F1 | PAA-β-D-GlcNAc / GlcNAc | No[c] | None | None |
| PBGlcNAc-GlcNAc8 | PAA-β-D-GlcNAc / GlcNAc | No[c] | None | None |

[a]Only sequenced phage clones that were cloned in pET-28a(+) for recombinant expression are mentioned here.
[b]Binding observed at high protein concentration.
[c]Recombinant protein could not be expressed.
[d]Recombinant protein was expressed but could not be purified.

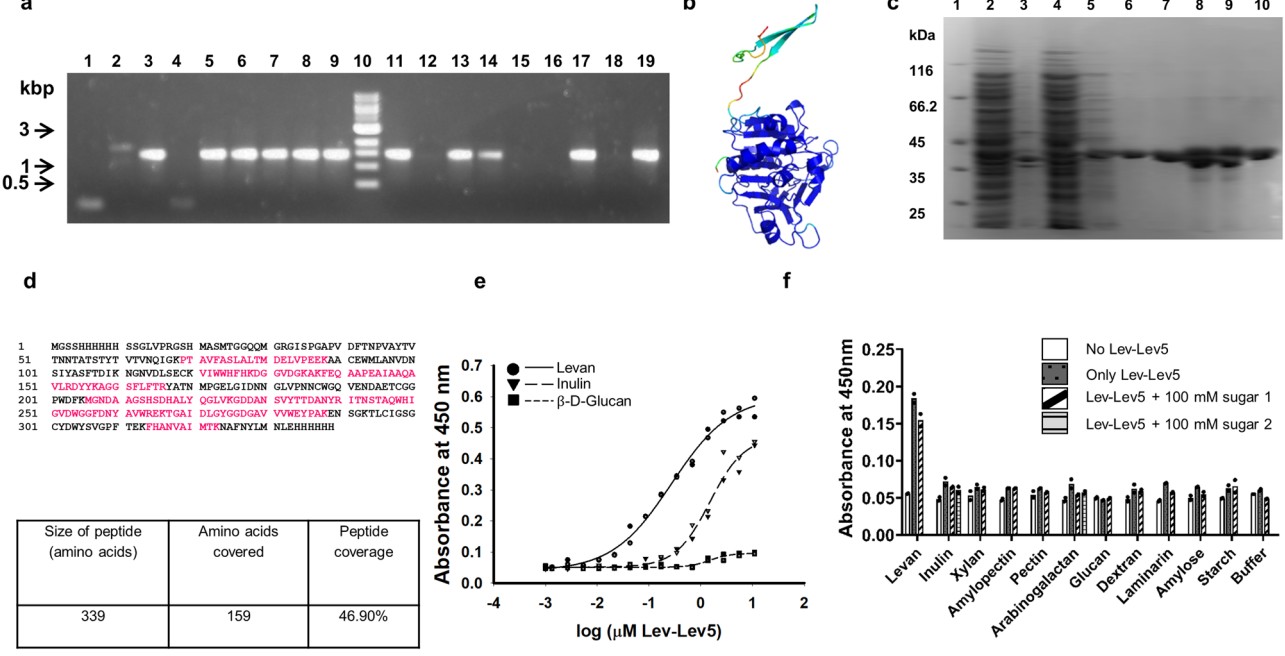

**Fig. 5 Identification of a levan binding protein domain. a** Agarose gel electrophoresis of PCR amplicons indicating the presence or absence of any recombinant insert following biopanning of the metagenomic phage display library against levan and elution with D-fructose or levan. Results of a single screening experiment. Lanes 1–9: PCR amplicons of clones Lev-Fru1 to Lev-Fru9. Lane 10: 1 kbp DNA marker. Lanes 11–19: PCR amplicons of clones Lev-Lev1 to Lev-Lev9. **b** AlphaFold2 predicted structural model of Lev-Lev5. **c** Coomassie blue stained SDS-PAGE showing purified Lev-Lev5 protein following Ni-NTA metal ion affinity chromatography. Lane 1: Protein molecular weight marker. Lane 2: Clarified lysate of *E. coli* BL21(DE3) expressing recombinant Lev-Lev5. Lane 3: Insoluble pellet fraction of *E. coli* BL21(DE3) expressing recombinant Lev-Lev5 following sonication. Lane 4: Flow through fraction during Ni-NTA metal ion affinity chromatography. Lane 5: 40 mM imidazole wash fraction during Ni-NTA metal ion affinity chromatography. Lanes 6–10: Purified Lev-Lev5 recombinant protein eluted with 250 mM imidazole during Ni-NTA metal ion affinity chromatography. **d** Sequence coverage obtained with MS/MS spectrometry of recombinant Lev-Lev5 protein followed by targeted sequence search; the peptide sequence coverage is shown in pink color. Results of a single mass spectrometry run. **e** Concentration dependent binding of Lev-Lev5 to immobilized levan, inulin and β-D-glucan. Results of a single binding experiment of two technical replicates. Individual values are plotted along with regression curves. Source data in Supplementary Data 3. **f** Binding of Lev-Lev5 to wells containing various immobilized carbohydrates with or without pre-incubation with component monosaccharide(s). The control used was wells without any immobilized carbohydrate. The polysaccharide carbohydrates with their monosaccharide units used in the assay are as follows. Levan: fructose. Inulin: D-glucose (sugar 1) and fructose (sugar 2). Xylan: xylose. Amylopectin: D-glucose. Pectin: D-galacturonic acid. Arabinogalactan: L-arabinose (sugar 1) and D-galactose (sugar 2). β-D-Glucan: D-glucose. Dextran: D-glucose. Laminarin: D-glucose. Amylose: D-glucose. Starch: D-glucose. Results of a single binding experiment of two technical replicates. Means are plotted as bars and individual data points included. Source data in Supplementary Data 3.

10–20 nM whereas the apparent affinity for immobilized inulin was ~10 fold higher (Fig. 5e, f, Table 2). Lev-Lev5 is therefore a novel levan binding protein domain.

**Identification of putative glucan, dextran, laminarin, pectin, and xylan binding protein domains.** We obtained the clones, BDG-Glc5 and BDG-BDG6 by biopanning against β-D-glucan and eluting with D-glucose and β-D-glucan, respectively (Fig. S8 and Supplementary Data 1, Table S4). We also obtained the clones Dex-Dex7/Lam-Glc9/Lam-Lam9 (eluted from dextran with dextran and also eluted from laminarin with glucose or laminarin), Pec-Pec1/Pec-GalA9 (eluted from pectin with galacturonate), and Xln-Xyl8 (eluted from xylan with xylose (Fig. S8 and Supplementary Data 1, Table S4).

Sequence analysis indicated similarity of BDG-Glc5 to β-mannanase and glycosyl hydrolase family 26 proteins from *Coprococcus* species that contained the following domains— GH26 (members of which have β-mannanase, exo-β−1,4-mannobiohydrolase, β−1,3-xylanase, endo-β−1,3-1,4-glucanase, and mannobiose-producing exo-β-mannanase activities as per CAZy), CBM11 (members of which bind to β−1,4-glucan and β−1,3/4 glucans as per CAZy), CBM27 (members of which bind to mannans as per CAZy), and complex I intermediate-associated protein 30 (a domain similar to CBM11 in Pfam-A HH search) (Supplementary Data 1, Table S5). Further, BDG-Glc5 itself contained CBM27 and (putative mannan binding) CBM23 domains in searches against the dbCAN2 database (Supplementary Data 1, Table S6). BDG-BDG6 showed similarity to *Prevotella copri* polygalacturonases and an IPT/TIG domain containing protein but showed no hits in searches against the dbCAN2 and Pfam databases (Supplementary Data 1, Tables S5, S6, S7, and S8).

Dex-Dex7 shared sequence similarity with proteins from *Prevotella* species, which contain pectin esterase, bacterial Ig-like and divergent InlB B-repeat domains (Supplementary Data 1, Table S5). Pec-Pec1 was also most similar to proteins from *Prevotella* species and *Bacteroidales* bacterium, which contain pectin esterase, bacterial Ig-like and divergent InlB B-repeat domains (Supplementary Data 1, Table S5), and an all against all protein sequence search showed that Dex-Dex7 and Pec-Pec1 shared 94% sequence identity, albeit over an alignment region of just 17 amino acids (Supplementary Data 1, Table S9). We found no hits in searches against the dbCAN2 and Pfam databases for Dex-Dex7 or Pec-Pec1 sequences (Supplementary Data 1, Tables S6, S7, S8).

Sequence searches indicated that Xln-Xyl8 was most similar to proteins from *Prevotella* species containing a GH10 domain and a carbohydrate binding domain (Supplementary Data 1, Table S5), and Xln-Xyl8 itself has a GH10 and a CBM4/CBM16, and a GH10 and a carbohydrate binding domain (CM4/CBM9/CBM16/CBM22) as per dbCAN2 and Pfam database searches, respectively (Supplementary Data 1, Tables S6, S7). AlphaFold2 predicted structural models indicate the presence of β-sandwich domains in all these proteins (Fig. S8).

Therefore, all these sequences (BDG-Glc5, BDG-BDG6, Dex-Dex7, Pec-Pec1, Xln-Xyl8) represent hitherto uncharacterized proteins involved in glycan binding and/or metabolism of glucan, dextran/laminarin, pectin, and xylan. Biochemical experiments are required to characterize the carbohydrate binding function in these proteins (Table 2).

**Phage display selection against inulin, arabinogalactan, peptidoglycans, and polyacrylamide-based glycoconjugates.** We obtained the clones, Iln-Glc6 (eluted from inulin with D-glucose), AG-Gal2 (eluted from arabinogalactan with D-galactose),

StapPG-GlcNAc6 (eluted from *Staphylococcus aureus* peptidoglycan with GlcNAc), StapPG-StapPG6 and StapPG-StapPG7 (eluted from *S. aureus* peptidoglycan with *S. aureus* peptidoglycan), MethPG-MethPG8 (eluted from *Methanobacterium* sp. peptidoglycan with *M.* sp. peptidoglycan), PBGal-Gal8 (eluted from PAA-β-D-Gal with D-galactose), and PBGlcNAc-GlcNAc4, PBGGlcNAc-GlcNAc8, PBGlcNAc-GlcNAc9 and PBGlcNAc-GlcNAc17 (eluted from PAA-β-D-GlcNAc with N-Acetyl-D-glucosamine) following eight rounds of biopanning (Fig. S9 and Supplementary Data 1, Table S4).

AG-Gal2, MethPG-MethPG8 and StapPG-GlcNAc6 showed similarity to translation initiation factor IF-2 proteins from *Prevotella* species (Supplementary Data 1, Table S5). Iln-Glc6 showed similarity to Ribonuclease from *Bacteroides* species and *Bacteriodales* bacterium, and PBGal-Gal8 showed similarity to Helix-hairpin-helix domain containing proteins from *Prevotella copri* (Supplementary Data 1, Table S5). StapPG-StapPG6 showed similarity to DEAD-DEAH box helicases from *Prevotella* species, and StapPG-StapPG7 showed similarity to N-acetylmuramoyl-L-alanine amidases from the gut metagenome and *Prevotella* species (Supplementary Data 1, Table S5). A Helicase-conserved C-terminal domain and a Helix-Hairpin-helix motif were found in PBGal-Gal8 and StapPG-StapPG6 by Pfam-A searches, and hits were returned for AG-Gal2, Iln-Glc6, MethPG-MethPG8, StapPG-GlcNAc6, StapPG-StapPG6, and PBGal-Gal8 by Pfam-B searches (Supplementary Data 1, Tables S7, S8). All against all protein sequence searches revealed similarity between MethPG-MethPG8, AG-Gal2, and StapPG-GlcNAc6 (Supplementary Data 1, Table S9). AlphaFold2 did not predict structures with good pLDDT scores for any of these protein sequences (Fig. S10). It was surprising that these clones showed similarity to proteins involved in transcription, translation, recombination, or repair and no similarity to CBMs or CAZymes. Further experiments are in order to confirm carbohydrate binding function in these proteins (Table 2).

PBGlcNAc-GlcNAc4, PBGlcNAc-GlcNAc8, PBGlcNAc-GlcNAc9, and PBGlcNAc-GlcNAc17 in their expected translation frames (listed as PBGlcNAc-GlcNAc4F1, PBGlcNAc-GlcNAc8, PBGlcNAc-GlcNAc9F1, and PBGlcNAc-GlcNAc17F1, respectively, in Supplementary Data 1, Table S4) did not find hits to any proteins in the non-redundant database or to any protein domain in searches against dbCAN2 and Pfam databases. However, the amino acid sequences obtained from the second frame of translation of PBGlcNAc-GlcNAc4, PBGlcNAc-GlcNAc9, and PBGlcNAc-GlcNAc17 (listed as PBGlcNAc-GlcNAc4F2, PBGlcNAc-GlcNAc9F2, and PBGlcNAc-GlcNAc17F2 in Supplementary Data 1, Table S4) shared sequence similarity with α−1,4-glucan phosphorylase, α-L-fucosidase, and glycosyl hydrolase family 2 proteins, respectively (Supplementary Data 1, Table S5). Further, dbCAN2 searches indicated the presence of GT35, GH29 and GH151, and GH106 domains in PBGlcNAc-GlcNAc4F2, PBGlcNAc-GlcNAc9F2, and PBGlcNAc-GlcNAc17F2, respectively. Pfam-A searches indicated the presence of carbohydrate phosphorylase domain in PBGlcNAc-GlcNAc4F2, and α-L-fucosidase and hypothetical glycosyl hydrolase 6 domains in PBGlcNAc-GlcNAc9F2, and Pfam-B searches returned hits for PBGlcNAc-GlcNAc4F2 and PBGlcNAc-GlcNAc17F2 (Supplementary Data 1, Tables S6, S7, S8). AlphaFold2 predicted structures of PBGlcNAc-GlcNAc4F2, PBGlcNAc-GlcNAc9F2, and PBGlcNAc-GlcNAc17F2 had good pLDDT scores (Fig. S10). We therefore suspect that these phage clones harbor a frameshift mutation in gene 10B upstream of the metagenomic insert cloning site, and that our screening has indeed resulted in enrichment of valid carbohydrate binding domains against PAA-β-D-GlcNAc, albeit we have not confirmed their carbohydrate binding function by biochemical assays (Table 2).

## Discussion

Carbohydrate-Active Enzymes (CAZymes) mediate the metabolism of complex glycans in the gut. The CAZy database[18], dbCAN2[33,38], and CAZymes Analysis Toolkit (CAT)[39] are popular databases used for CAZyme prediction but their predictive ability of substrate specificities is low due to lack of structurally and biochemically characterized members with regard to the ever increasing volume of (genomic and) metagenomic sequence data. Further, sequence based metagenomic analysis can only assist in the annotation of new sequences that are similar in sequence (evolutionarily related) to existing characterized sequences. It cannot aid in the discovery of new protein families with a desired function.

Previous function-based screening studies have demonstrated that complex microbial communities are a rich treasure trove for the discovery of new sequence classes of novel functionalities[40–44]. Fosmid libraries have been used to mine CAZymes involved in plant polysaccharide degradation from human[45] and wood-feeding termite[46] gut metagenomes, and metagenomic phage display libraries have been employed for the discovery of potential disease biomarkers[47], analysis of the metasecretome component of rumen microbial community[48], and for the analysis of IgA and fibronectin binding proteins from the human tongue dorsum[49]. However, the human gut microbiome remains vastly under-explored with regard to the glycan binding proteins present within it. Our study has extended this metagenomic exploration to the efficient screening of glycan binding protein sequences, and revealed the existence of several previously unknown protein domains—mucin binding domains (MG1, MN3, MU1, and MU3), a β-D-glucan binding domain (BDG-BDG6), a dextran and laminarin binding domain (Dex-Dex7/Lam-Glc9/Lam-Lam9), and a pectin-binding domain (Pec-Pec1). Further, the identification of relevant carbohydrate binding protein domains such as CBM26[50] and Outer membrane protein SusF_SusE domain[51,52] in our starch binding screen, CBM4/CBM16 in our xylan binding screen, and CBM23 and CBM27 in our β-D-glucan binding screen, is an encouraging sign for the use of functional metagenomics screens for the discovery of glycan binding proteins.

Mucins are highly O-glycosylated proteins with glycosylation contributing to >50% of their molecular mass[12] and a rich oligosaccharide repertoire of ~100 different O-glycan structures that are differentially present along the length of the gut, thereby creating unique niches and potential binding sites along the gut for the colonization of bacteria[11,53], perhaps contributing to the varying microbiota composition in different regions of the gut[54,55]. The ability to bind to or utilize mucin provides a competitive edge in colonization of the gut[56–59], and gut symbionts like *Akkermansia muciniphila*[60], *Bifidobacterium*[61], *Bacteroides* species[58], and *Lactobacillus fermentum*[62] indulge in mucosal glycan foraging, secreting glycoside hydrolases (GHs) for mucin glycan degradation[59,63].

We found that MG1, MN3, and MU1 preferentially bind to terminal, non-sialylated N-acetyllactosamine (LacNAc) moieties of complex N-glycans with fine variations whereas MU3 showed strict recognition of α−2,6-linked sialylated complex N-glycans. Binding to glycans was weak with dissociation constants in the high micromolar to low millimolar range, and binding to several other glycans was not detected, indicating that these proteins bound weakly to glycans and had been enriched and identified in the metagenomic screen thanks to the avidity provided by the multivalent presentation during both ligand immobilization and phage display. Considering the O-glycan rich nature of mucins, it is surprising that all four mucin-binding proteins identified in this study bind to N-glycan structures. However, it is important to note that our conclusions of glycan binding specificity are based on the mammalian glycan array v5.2, which has a lower representation of O-glycans with lactosamine extensions.

Based on our biochemical evidence of glycan binding (and the high sequence similarity between MU1 and MU3), we suggest that MG1, MN3 and MU1/MU3 domains be annotated as new CBMs in the CAZy database. Future studies can focus on how these glycan binding domains compare with existing LacNAc binding galectins[21,64] and Siaα2-6 binding proteins such as the human influenza A and B virus lectin haemagglutinin[65], *Pseudomonas aeruginosa* pili[66], the surface protein antigen (PAc) of *Streptococcus mutans*[67], mushroom *Polyporus squamosus* agglutinin (PSA)[68], and the plant lectins, *Sambucus nigra* agglutinin (SNA), *Sambucus canadensis* agglutinin (SCA), *Sambucus sieboldiana* agglutinin (SSA), Wheat-germ agglutinin (WGA), *Trichosanthes japonica* (TJA-I), and ML-I of *Viscum album*[69–73].

We identified Lev-Lev5 (DUF4960) as a novel levan binding protein with a Rossmanoid flavodoxin-like fold, with higher affinity for levan (fructose polymer in bacteria[74] and some plants with a β−2,6 linked backbone and β−2,1 branches) than inulin (fructose polymer in plants with β−2,1 linkages). Analysis of the domain architectures of DUF4960 in Pfam indicates that most (84 out of 103) sequences listed have a large unannotated N-terminal region in the polypeptide, hinting at the existence of an as yet, unannotated protein domain, perhaps involved in levan/inulin metabolism. Interestingly, there are a few (5 out of 103) DUF4960 sequences listed in Pfam that also have co-occurring (fructan metabolizing) GH32 N- and C-terminal domains. To our knowledge, only two inulin/levan binding protein domains have been reported in literature—(1) the non-catalytic CBM38 domain (previously named X39 module; residues 32-389) of *Paenibacillus macerans* (*Bacillus macerans*) GH32 cycloinulooligosaccharide fructanotransferase (GenBank: AAG47946.1), which comprises two β-sandwich domains (AlphaFold2 predicted structure in Fig. S11) and binds to levan/inulin in the low micromolar range[75], and (2) the non-catalytic CBM66 β-sandwich domain of *Bacillus subtilis* exo-acting β-fructosidase SacC, another GH32 enzyme[19], which demonstrated binding to the terminal non-reducing end of levan. Based on our biochemical evidence for glycan binding in Lev-Lev5 and its novel protein fold, we suggest that DUF4960 be annotated as a new CBM family in the CAZy database.

Based on the contextual evidence of protein sequence/ domain architecture, the absence of similarity with known CBM families, and the high sequence similarity between Dex-Dex7/Lam-Glc9/ Lam-Lam9 and Pec-Pec1, we propose that BDG-BDG6 and Dex-Dex7/Lam-Glc9/Lam-Lam9/Pec-Pec1 be annotated as putative CBMs. It is interesting that dextran (α−1,6 linked glucose with α −1,3 branches), laminarin (β−1,3 linked glucose with β−1,6 branches), and pectin (α−1,4-linked galacturonic acid) are recognized by the same/similar protein domains. Future studies may compare the carbohydrate binding of these domains with that of CBMs already annotated to bind to β-D-glucans (β−1,3/4 linked glucose)—CBM4, CBM6, CBM11, CBM22, CBM28, CBM43, CBM52, CBM65, CBM72, CBM76, CBM78, CBM79, CBM80, and CBM81, and pectin—CBM7, respectively.

Overall, most of the sequenced clones identified in our screen shared highest similarity with proteins of the genus *Prevotella*. This is likely because *Prevotella* is the most abundant genus in the feces of adult Indian subjects[76,77]. *Prevotella* has been associated with long-term diets rich in carbohydrates[78] and Indian societies are mainly agrarian, deriving most of their nutrition from plant-based products. *Prevotella* sequences might thus have had an advantage in outcompeting other sequences during the many rounds of biopanning performed in our study. Further, we had performed eight rounds of biopanning (alternating blocking reagents after every two rounds of biopanning to eliminate clones binding to the blocking reagent) to avoid false positives[28], considering the phage titres obtained following 3–4 rounds. It is

possible that we have missed carbohydrate-binding protein domains from less represented bacterial species due to this. Future screening efforts involving DNA subtraction[79] together with the use of sample DNA from different body sites might help capture more novel carbohydrate binding proteins and domains from underrepresented species.

To conclude, our work has described the glycan binding specificities of several new and previously unannotated microbial carbohydrate binding protein domains identified through a metagenomic phage display screen, and provided the prima facie evidence for the carbohydrate binding function and glycan binding specificities of some of these domains. Detailed future studies will be useful to assess their potential in lectin and CBM-related applications. Future studies could also focus on the catalytic domains co-occurring with these new CBM families in proteins. In-depth studies of the gut microbial glycan binding proteins and their interactions with mucin in the host intestinal epithelium might also offer insights into the dynamic as well as resilient nature of the microbial species in our gut and their remarkable influence on our health.

## Methods

**Purification of T7Select 10-3b phages**. The T7Select 10-3b phage vector (Novagen, Merck) was used in this study. Non-recombinant/recombinant T7Select 10-3b phages were purified from the lysate of infected *E. coli* strain BL5403 by PEG-8000 precipitation as per the method described in T7 select system manual. The pellet was suspended in sterile water, to which equal volume of chloroform was added, vortexed to mix well, and centrifuged for 5 min at 5000 rpm. The aqueous layer containing phages was pipetted out avoiding the interface, and passed through a 0.22 μm filter in a sterile tube to obtain purified phages. Finally, phage DNA was purified by phenol-chloroform extraction and ethanol precipitation.

**Preparation of packaging extract**. We made the packaging extract with the lysate of replication-deficient T7 Δ9-10B, D104, Δ38 phage, following the protocol as kindly provided by Prof. F. William Studier, Brookhaven National Laboratory and Dr. Tatjana Heinrich, Institute for Child Health Research, Western Australia.

For the propagation of replication defective phage, an overnight culture of *E. coli* BL21/pAR3924,5453 was set up by inoculation of a single colony in 5 ml of LB/ G medium (LB + 0.3% filter sterile glucose with 50 μg/ml Carbenicillin and 30 μg/ ml Chloramphenicol), and the culture was grown with shaking at 200 rpm at 37 °C. A secondary culture was set up by diluting the overnight culture 1:200 into 50 ml of LB/G medium and grown with shaking at 200 rpm and 37 °C until the cell density corresponded to an OD$_{600}$ of 0.5. Then, 5 ml of this culture was transferred to a tube and grown for another 30 min and kept at 4 °C till required. The remaining 45 ml culture was infected with 10 μl of replication defective phage (PEG-precipitated, titer: 1–2 ×10[11] pfu/ml) and kept in an incubator shaker at 37 °C for 3–4 h until lysis was observed. Lysis was determined by comparing the O.D. of culture with and without phage addition. Following lysis, the cell debris was pelleted down by centrifugation at 12,000 rpm for 30 min at 4 °C and the supernatant was transferred into a fresh tube. For long-term storage, the clarified lysate was kept at −80 °C without addition of glycerol. For the generation of packaging extract, phage-containing lysate was PEG-precipitated by the addition of 0.1 volume of 5 M NaCl and 0.166 volume of 50% PEG-8000, mixing well on a vortex, and incubation at 4 °C overnight. Subsequently, the mixture was centrifuged for 20 min at 4 °C at 12,000 rpm. The supernatant was removed carefully, and the pellet was briefly air-dried and resuspended in PBS (one-tenth the volume of original phage-containing lysate). The titer was checked with *E. coli* BL21/pAR3924,5453 culture as described in T7Select System (Novagen) manual.

For the propagation of replication defective phage for T7 packaging extract, an overnight culture of BL24/pAR3924,5453 host strain was set up in medium (LB + 0.3% filter sterile glucose with 50 μg/ml Carbenicillin and 30 μg/ml Chloramphenicol) by inoculation of one colony into 5 ml of broth and the culture was grown with shaking overnight at 37 °C. Then, 200 ml of a secondary culture was set up by diluting the overnight culture 1:200 in a 1 liter flask containing LB/G medium and culturing with continuous shaking at 200 rpm and 30 °C until the cell density corresponded to an OD$_{600}$ of 1. At OD$_{600}$ of 1.0, the bacterial culture was infected with replication defective phage at multiplicity of infection (MOI) of 7 and grown by shaking (200 rpm) at 30 °C. Exactly after 25 min of infection, the culture was chilled by placing the flask in ice water for ~5 min. Then, the culture was transferred into a cold centrifuge bottle and cells were pelleted for 2 min at 8000 rpm at 4 °C, and the remaining procedure was performed at 4 °C. The supernatant was decanted and removed, and the pellet was resuspended in 25 ml of cold extract buffer (100 mM NaCl, 20 mM Tris-Cl pH 8.0, 6 mM MgSO$_4$). Cells were pelleted again at 8000 rpm for 2 min and the supernatant was removed. The pellet was resuspended in 400 μl of extract buffer using a pipette tip, and

transferred to a cold microcentrifuge tube kept on ice. The cells were lysed by freezing in liquid nitrogen for 5 min and thawed to 30 °C quickly in water bath. The freeze-thaw was repeated, and the tube placed on ice for 5 min, and the lysate cleared by centrifugation at 12,000 rpm at 4 °C for 10 min. The supernatant (clarified extract) obtained was saved. To make packaging extract, 25 μl of 50% dextran, 3.33 μl each of MgSO$_4$, ATP, and Spermidine, and 2 μl of β-mercaptoethanol (diluted 1:100 volume/volume in water) were added per 100 μl of clarified extract. Then, the packaging extract was aliquoted and stored at −80 °C.

In vitro T7 phage packaging was done according to the T7Select System (Novagen) manual. The number of recombinants generated was determined by performing a plaque assay as described in the manual.

**Preparation of non-recombinant T7Select 10-3b phages**. To generate non-recombinant phages (henceforth referred to as T7Select 10-3b phage), 9 μl packaging extract, 1 μl T7Select 10-3b DNA (100–500 ng of T7 DNA), and 1 μl T7 DNA polymerase (NEB; 10 U/μl) diluted 1:10 in extract buffer were mixed and incubated at 22 °C for 2 h. The reaction was stopped by the addition of 90 μl of LB. The titer was checked as per instructions in the Novagen manual.

**Construction of recombinant fucose-binding T7Select 10-3b phage (*Sr*NaFLD-T7Select 10-3b phage)**. A 759 bp fragment corresponding to the Na domain and the fucose binding F-type lectin domain (FLD) of a *Streptosporangium roseum* gene,*Sr*FucNaFLD (GenBank accession no: ACZ87343.1) was PCR amplified from an *Sr*FucNaFLD-pET-28a(+) construct (Bishnoi, R. et al.) using Taq DNA polymerase, forward primer with an EcoRI site: (5′-CCG GAA TTC TGT CCG TTC ACT GCT GCG CAC C-3′) and reverse primer with a HindIII site: (5′-CCC AAG CTT GCC ACG CAC TTG GAC TTC TGC CA−3′).

The EcoRI and HindIII digested gene fragment was ligated to the EcoRI and HindIII digested T7 vector arms (T7Select System, Novagen) according to the user manual. Briefly, for sticky end ligation, 0.5 μl (0.5 μg/μl) of EcoRI and HindIII digested vector arms and EcoRI and HindIII digested insert in a 1:3 (vector:insert) ratio were assembled together with 0.5 μl 10× T4 ligase buffer, 0.5 μl 10 mM ATP, 0.5 μl 100 mM DTT, 1 μl T4 DNA ligase and water in a total reaction volume of 5 μl and incubated at 16 °C for 16 h.

The ligation mixture was packaged into the packaging extract, the phages were amplified and plated. Plaque plugs were picked by micropipette tips and placed in 30 μl of Phage Extraction Buffer (20 mM Tris-HCl, pH 8.0, 100 mM NaCl, 6 mM MgSO$_4$). They were screened by PCR amplification using T7SelectUP (5′-GGAGC TGTCGTATTCCAGTC-3′) and T7SelectDOWN (5′-AACCCCTCAAGACCCG TTTA-3′) primers for the presence of insert. Recombinant phages (referred to as *Sr*NaFLD-T7 phages) from single plaques obtained from agar plugs were amplified in fresh culture to rule out the possibility of the presence of non-recombinant phages in the stock.

**Construction of metagenomic phage display library**. This study was approved by Council of Scientific and Industrial Research (CSIR)- Institute of Microbial Technology Institutional Ethics Committee (Human) (Project number 11 IEC/1/9-2014) and Council of Scientific and Industrial Research (CSIR)- Institute of Microbial Technology Institutional Biosafety Committee (Project number IBSC/ 2012-2/21). We received written informed consent to release the information obtained and publish the study following the subject's participation without disclosing the subject's identity. Subjects were self-reported healthy individuals with local dietary intake from Chandigarh (30.7333˚N, 76.7794˚E), Ladakh (34.425960˚N 76.824421˚E), Khargone (22.226704˚N, 75.863329˚E), and Jaisalmer (26.36539˚N, 70.42584˚E) in the age group of 18–58 years. Eleven of the 50 samples were from female subjects, and the mean age was 32. Samples from Ladakh, Khargone, and Jaisalmer were collected on 27 October 2013, 25–26 November 2014, and 9 November 2014, respectively. Fecal samples were collected in sterile containers, transported on dry ice, and stored at −80 °C until further processing.

Metagenomic DNA isolation was done as follows. DNA was isolated from 50 fecal samples by 45 volunteers using ZR Fecal DNA MiniPrep™ (for 18 samples), MoBio PowerFecal DNA Isolation Kit (for 27 samples), QIAamp DNA stool minikit (for 4 samples), and MDI Stool Genomic DNA Miniprep Kit (for 1 sample) following the respective manufacturers' instructions. The procedure was slightly modified for isolation of DNA using ZR Fecal DNA MiniPrep™. Lysis buffer was added to (up to) 150 mg of fecal sample and the sample was subjected to three rounds of homogenization (speed 6) for 40 s each (with cooling on ice for 1 minute after each round of homogenization) using FastPrep 120. The remaining steps of the protocol were performed as per the manufacturer's instructions. Isolated DNA was subjected to RNAse treatment followed by phenol-chloroform extraction and ethanol precipitation. The quality of the isolated DNA was assessed by visualization using agarose gel electrophoresis and the DNA was quantified by using NanoDrop 1000 spectrophotometer.

Metagenomic DNA fragmentation and repair was done as follows. The DNA isolated from different fecal samples was fragmented by sonication to generate maximum fragments of 500–3000 bp. Fragmented DNA of 500–3000 bp was resolved by electrophoresis on a 1% agarose gel, excised, and purified by QIAquick Gel Extraction Kit. Equal amounts of purified DNA from all the samples were

pooled. This pooled DNA was end repaired using NEBNext End Repair Module (New England BioLabs).

T7 phage vector digestion and dephosphorylation was done as follows. T7 phage vector arms were prepared by restriction digestion with SmaI (New England Labs) at 25 °C for 4 h to generate blunt ends, followed by heat inactivation of SmaI at 65 °C for 20 min, and dephosphorylation by the addition of Calf Intestinal Alkaline Phosphatase (New England Labs) and incubation at 37 °C for 30 min. Vector arms were purified by phenol extraction.

For ligation of T7 phage vector arms and metagenomic DNA fragments, the following procedure was used. A blunt end ligation reaction was set up by assembling 1 μl (1 μg/μl) SmaI digested vector arms, and end repaired 500–3000 bp metagenomic DNA in a 1:6 (vector arms: insert) ratio (considering 1500 bp as average size), 1 μl 10x T4 ligase buffer, 1 μl 100 mM ATP, 1 μl 100 mM DTT, 2 μl 50% PEG-4000, 2 μl of T4 ligase and TE buffer in a total reaction volume of 10 μl. The ligation mixture was then incubated at 16 °C for 24 h in a water bath.

For packaging and amplification of the metagenomic phage display library, the following procedure was used. The ligation mixture was packaged into the packaging extract to obtain a human fecal metagenomic phage display library (of 6 ml volume) with a titer of $2 \times 10^6$ pfu/ml and a total of $1.2 \times 10^7$ phages. The phages were amplified and the titer of the amplified metagenomic phage display library was $3 \times 10^{11}$ pfu/ml. The amplified metagenomic phage display library therefore contained ~$3 \times 10^4$ copies of each recombinant phage clone per ml of metagenomic phage display library. The amplified metagenomic phage display library was plated, and recombinant phage plaques screened by PCR amplification using T7SelectUP (5′-GGAGCTGTCGTATTCCAGTC-3′) and T7SelectDOWN (5′-AACCCCTCAAGACCCGTTTA-3′) primers for the presence of insert.

### Preparation of 96-well plates coated with carbohydrate ligands for biopanning.
Mucin from porcine stomach (Sigma) dissolved in sodium carbonate-bicarbonate buffer (pH 9.2) was applied (100 μl of 100 μg/ml each well) to 96-well plates (Nunc MaxiSorp ELISA plates) overnight at 4 °C. Each well was washed twice with 200 μl phosphate-buffered saline (8 mM $Na_2HPO_4$, 150 mM NaCl, 2 mM $KH_2PO_4$, 3 mM KCl, pH 7.4) containing 0.05% Tween 20 (PBS-T) and once with 200 μl PBS. Then, the plates were blocked with blocking agents for 4 h at room temperature. Bovine Serum Albumin (BSA) (300 μl of 3% in PBS) was used for blocking in the first, second, fifth, and sixth cycles of phage display selection whereas skim milk powder (300 μl of 5% in PBS) was used for the third, fourth, seventh, and eighth selection cycles. The plates were washed thrice with PBST and twice with PBS. Finally, 100 μl of water was pipetted to each well, sealed with parafilm and stored at 4 °C.

For Biotin-PAA-sugars, NeutrAvidin-coated plates were made by incubating 50 μl (4 mg/ml in deionized, sterile water) of NeutrAvidin in 96-well plates (Nunc MaxiSorp ELISA plates) overnight at 4 °C. The remaining steps were similar to that in mucin plate preparation except for the composition of the buffers used - TBS (20 mM Tris, 150 mM NaCl, pH 7.4) and TBS-T (TBS + Tween-20 0.1%). Before incubation with phages (initiating biopanning) NeutrAvidin coated plates were incubated with biotin-PAA-sugars (Glycotech; 50 μl of 10 μg/ml) (listed in Table S1) for 1 h at 37 °C. Washing was done thrice with TBS-T and twice with TBS.

Water soluble polysaccharides and peptidoglycans (Sigma; listed in Table S1) were applied (100 μl of 10 μg/ml in deionized, sterile water) to 96-well plates (Costar 3598 plates) followed by drying to the well surface by evaporation overnight at 37 °C[24]. Water insoluble polysaccharides were dissolved in DMSO (1 mg/ml) and 1 μl of the solution was applied to each well followed by 99 μl of deionized water to Costar 3598 plates. Then, it was left for drying by evaporation overnight at 37 °C. The remaining steps of biopanning were as mentioned for mucin and NeutrAvidin. The buffers used were TBS and TBS-T.

### Biopanning.
Ligand-coated plates were incubated with 100 μl of phages (overnight at 4 °C in first selection cycle, for 2 h at 37 °C in selection cycles 2–5, and for 2 h at 37 °C with shaking at 100 rpm in selection cycles 6–8), washed 6 times with PBST (in case of mucin) or TBST, and thrice with PBS (in case of mucin) or TBS, and then incubated with the appropriate elution reagent (as listed in Table S1 for other ligands) for 1 h at 37 °C to elute bound phages. After 1 h of incubation at 37 °C, the elution reagent containing phages was pipetted out into sterile microcentrifuge tubes. Next, 10 μl of the elution reagent was used for checking titer and the remaining solution was used for amplification in host cell culture. Following lysis of host cells, the debris was pelleted by centrifugation at 12,000 rpm for 15 min at 4 °C. The supernatant was used for the next cycle of phage display selection. Eight cycles of phage display selection were performed.

For biopanning of the metagenomic phage display library, we calculated the multiplicity of screening to be ~3000, considering that we used 100 μl of amplified metagenomic phage display library with a titer of $3 \times 10^{11}$ pfu/ml (which contained ~$3 \times 10^4$ copies of each recombinant phage clone per ml of metagenomic phage display library).

### Sequencing and sequence/structure analysis.
Following selection, randomly picked phage plaques were screened by PCR amplification using T7SelectUP (5′-GGAGCTGTCGTATTCCAGTC-3′) and T7SelectDOWN (5′-AACCCCTCAAGACCCGTTTA-3′) primers for the presence of insert. Phage plaques with amplicon size >144 bp (which is the size of the amplicon expected for a non-recombinant phage amplified by the T7UP and T7DOWN primers) were considered to be recombinant. Enriched recombinant phages were further subjected to Sanger sequencing in-house (at IMTECH, using 16-capillary 3130xl Genetic Analyzer, Applied Biosystems). DNA sequences obtained were translated using ExPASy[80] and visualized using FinchTV version 1.4.0. Sequences with more than 30 continuous amino acids (without any stop codon) were considered for further analysis. These metagenomic sequences were subjected to searches against both the protein sequence database, 'UniRef100'[81], and the profile databases, 'pfamA-full' and 'pfamB'[31], and 'dbCAN2'[33] using mmseqs2[82] at a very high sensitivity (-s 8.5). In addition, an all-against-all sequence search of the initial 33 query proteins was also performed. The general command run for these searches was 'mmseqs easy-search input_protein_seqs search_database output.result_file tmp_directory -s 8.5 --format-mode 2'. In the search against UniRef100, for each query protein sequence, the top five returned hits were retained and were then further searched for pfamA-full domains as above using mmseqs2. The fields 'representativeMember_organismName' and 'cluster_representative_name' (in Table S5) were obtained using uniprot API[81]. In the search result against pfamA-full, the field 'pfam-family_description' (in Table S7) was obtained using https://github.com/AlbertoBoldrini/python-pfam (a python interface for pfamA). Three-dimensional structure predictions were performed for all amino acid sequences using a local server installed AlphaFold2, and structures were visualized using reverse rainbow coloring based on pLDDT scores in PyMOL (Schrodinger) (Supplementary Data 2).

### Cloning, expression, and purification of recombinant metagenomic insert sequences.
The PCR amplicons of the clones, MG1 and MN3, amplified with the primers, forward: 5′-GATCCGAATTCTCTCCTGCAGGGATATC-3′ and reverse: 5′-AAC CCC TCA AGA CCC GTT TAG AGG CC-3′, were digested with the restriction enzymes, EcoRI and HindIII. Similarly, the PCR amplicons of the clones, MU1 and MU3, amplified with the primers, forward: 5′-CAGCCATATG GGAGCTGTCGTATTCCAGTC-3′ and reverse: 5′-AAC CCC TCA AGA CCC GTT TAG AGG CC-3′, were digested with the restriction enzymes, NdeI and XhoI. These PCR amplicons were ligated into the expression vector, pET-28a(+), digested with corresponding enzymes and treated with CIP. E. coli TOP 10 cells were transformed with the ligation mixture and the transformants were grown on LB agar plates containing kanamycin (50 μg/ml) at 37 °C for 14-16 h. Transformant colonies were screened by restriction digestion and confirmed by DNA sequencing.

For expression of recombinant proteins, a primary overnight culture of E. coli BL21(DE3) cells transformed with the pET-28a(+) plasmid clone was used to inoculate LB medium containing kanamycin (50 μg/ml). The cultures were grown with continuous shaking at 200 rpm to an $OD_{600}$ of 0.6–0.8 at 37 °C, whereupon recombinant protein expression was induced as follows. We used 0.1 mM IPTG for 3 h 30 min at 37 °C with 200 rpm shaking for the expression of MG1, MN3, MU1, and MU3 protein (subsequently used for glycan array analysis). We used 0.1 mM IPTG for 10 h at 22 °C with 180 rpm shaking for the expression of MG1 (subsequently used for CD spectroscopy and ITC assays). We used 1 mM IPTG for 3 h 30 min at 37 °C with 200 rpm shaking for the expression of MN3, MU1, and MU3 (subsequently used for CD spectroscopy, ELLA and ITC assays). We used 1 mM IPTG for 3 h at 37 °C with 200 rpm shaking for the expression of Lev3, St1, and St5.

For protein purification, cells were harvested by centrifugation at $4000 \times g$ for 7–10 min. Recombinant proteins with N-terminal hexahistidine tags were purified by metal ion affinity chromatography. The cell pellet was resuspended in lysis buffer, the composition of which was as follows. We used 150 mM or 300 mM NaCl, 25 mM imidazole, and 20 mM Tris, pH 7.4 for MG1, MN3, MU1, and MU3. For MG1, where sonication was used to lyse the cells, the lysis buffer also included 20 mU/mL DNase I (Roche), 100 μg/mL lysozyme (Sigma) and 1 mM PMSF (Sigma). We used 150 mM NaCl, 20 mM Tris, pH 7.5 for Lev-Lev5. We used 150 mM NaCl, 0.5% N-lauryl sarcosine and 20 mM Tris, pH 7.5 for St-Glc1 and St-Glc5v2.

The cells were disrupted using a French Press (using 300 mM NaCl; in case of MG1, MN3, MU1 and MU3 proteins subsequently used in glycan array analysis) or a probe type ultrasonicator (Materials and Sonics INC) for 30 min (amplitude 25%, pulse 10 s on and 10 s off) (using 150 mM NaCl; for proteins expressed for all other biochemical assays), and the lysate was clarified by centrifugation at $16,000 \times g$ for 20–60 min at 4 °C. The supernatant was loaded on to a His-bind (Pierce metal affinity Ni-NTA resin) column (pre-equilibrated with lysis buffer) and incubated for 2–8 h at 4 °C with end-over-end rotation on a Rotospin (Tarsons). The column was then washed with wash buffer (40 mM imidazole in lysis buffer), and the protein eluted with elution buffer (250 mM imidazole in lysis buffer). The protein was subsequently dialyzed extensively against TBS, and the purity of the preparation evaluated by SDS-PAGE.

Where required (for MG1, MN3, MU1, and MU3), size exclusion chromatography was done using HiPrepTM 26/60 HR Sephacryl S-200 (GE Healthcare life sciences) column on a GE healthcare Akta Purifier Fast protein liquid chromatography (FPLC). For size exclusion chromatography, the duly washed column (with degassed sterile water) was equilibrated with 150 mM NaCl and 20 mM Tris, pH 7.5 at a suitable flow rate mostly (1.0 ml/min). The absorbance of the sample was measured at 280 nm and 260 nm. Peak fractions were collected

and the concentrated sample was assessed for purity by SDS-PAGE. Protein concentration was estimated by measuring the absorbance at 280 nm using Nano-Drop spectrophotometer.

Western analysis was performed using mouse anti-polyHistidine antibody (H1029, Sigma) and mouse anti-C-terminal-His antibody (R930-25, Invitrogen) followed by HRP-conjugated donkey anti-mouse IgG (715-035-150, Jackson Immunoresearch), mass spectrometry for intact mass analysis was performed on an Agilent i6550 Quadrupole Time Of Flight instrument with electrospray ionization (CSIR-IMTECH mass spectrometry facility) and in-gel trypsin digestion and MS/MS analysis was performed on a Thermo Fisher Scientific LTQ Orbitrap Velos Pro ion-trap mass spectrometer (Taplin Mass Spectrometry Facility, Harvard Medical School, Boston, Massachusetts, USA).

**Glycan micro-array analysis.** For analysis of glycan binding specificities, glycan microarray analysis of the purified recombinant proteins, MG1, MN3, MU1, and MU3 (in TBS with 10 mM CaCl$_2$) was performed at the National Center for Functional Glycomics (NCFG) at Beth Israel Deaconess Medical Center, Harvard Medical School using CFG glycan microarray version 5.3. Protein concentrations used were 5 µg/ml, 50 µg/ml, and 400 µg/ml for MG1, and 5 µg/ml and 50 µg/ml for MN3, MU1 and MU3. Briefly, the proteins were allowed to bind to the glycans in the microarray, the slide washed to remove non-specifically bound protein, and the bound protein detected with mouse anti-C-terminal 6xHis antibody and Alexa488-conjugated secondary anti-mouse antibody, followed by fluorescence scanning on a Perkin Elmer Scan Array Scanner. The spots with highest and lowest fluorescence intensity (of the eight spots for each glycan) were removed, and the remaining glycan array data analyzed by MotifFinder version 2.2.5[36,37]. We employed the automated model building function and default model building settings of MotifFinder (motif complexity parameter: 0.0, model complexity parameter: 0.01, minimum split size: 3, rebuild Fold-N times: 0, cross validation Fold N: 5, optimize top N leads per split: 2, test top N relations per split: 1, guide glycan selection seed: 0). With these settings, we used the glycan array data from multiple protein concentrations to build one model each for MG1, MN3, MU1, and MU3 (Fig. S4). Subsequently, we also used all the glycan array data of MG1, MN3, MU1, and MU3 together, and built a single automated model, altering the settings (motif complexity parameter: 0, model complexity parameter: 0.05, minimum split size: 5) to achieve a final list of binding motifs that was representative of the binding motifs of each of these proteins. Using this output, we then compiled a custom motif list, and used the manual model building function to build one model each for MG1, MN3, MU1, and MU3, using the glycan array data of all protein concentrations (Fig. 3).

**Circular dichroism spectroscopy.** Circular dichroism spectroscopic measurement for proteins MG1, MN3, MU1 or MU3 was done using a Jasco J815-1505 Spectropolarimeter (Jasco International co. ltd.). Spectra of protein solutions (0.2 mg/ml) in 20 mM phosphate buffer pH 7.4 were recorded at far UV (250-195 nm) region at a scan speed of 50 nm/min, a slit width of 1 nm, a data pitch of 0.1 nm, with three accumulations, and a path length of 1 mm. The mean residue molecular mass and number of amino acids in the protein sequence were used to calculate mean residue ellipticity, θ. The spectra plotted are moving averages of mean residue ellipticity calculated over a window of 10 readings.

**Isothermal calorimetry.** Isothermal calorimetry experiments were performed on a VP-ITC microcalorimeter (Malvern). The purified proteins MG1, MN3, MU1, and MU3, were extensively dialysed against suitable buffer, which was mostly 20 mM HEPES (4-(2-hydroxyethyl)−1-piperazineethanesulfonic acid), 150 mM NaCl, pH 7.5. For titrations of MG1, MN3, MU1, and MU3 against Lewis A tetraose, Lewis B tetraose, Lewis X tetraose, Lacto-N-difucohexaose II, and H antigen type-2 heptaose azido ethyl, and for titrations of MG1 and MU3 against H antigen type-2 pentaose β-N-Acetyl Propargyl, 20 mM phosphate buffer, pH 7.5 with 150 mM NaCl was used. The ligand solutions were prepared in the dialysate buffer. The experiments were performed by placing the protein in the sample cell and titrating ligand in a fixed volume at intervals of 180 s for 20 or 35 injections. To minimize any artifact associated with the loading of the ligand filled syringe, the first injection was performed with the volume of 0.2/0.4 µl and the corresponding data point was removed before final curve fitting. The heat changes due to heat of dilution were determined in two ways, one by titrating protein with buffer and another by titrating buffer with ligand. The experimental curve data was subtracted from one of these curves prior to data analysis. The subtracted data was fitted to a model using Microcal origin 7 analysis software and the thermodynamic parameters K, n and ΔH along with chi square value were obtained.

**Enzyme-linked lectin assays.** For MN3 binding assay, purified recombinant protein MN3 (100 µl of 50 µg/ml stock in TBS) was coated on a Maxisorp 96-well flat-bottom plate (Nunc) overnight at 4 ºC. Subsequently, the wells were washed twice with 200 µl TBST and once with TBS, blocked with 3% BSA for 4 h at room temperature, washed, incubated with 1 µg biotin-PAA-β-D-galactose, 1 µg biotin-PAA-α-D-galactose, 1 µg biotin-PAA-α-L-fucose, 1 µg biotin-PAA-β-D-GlcNac, 1 µg biotin-PAA-α-NeuAc, 1 µg biotin-PAA-Lewis$^y$, 1 µg biotin-PAA-H antigen type-2 triose, 1 µg biotin-PAA-α-D-Mannose, 1 µg biotin-PAA-β-D-Mannose or 1 µg biotin-PAA for 1.5 hours at 37 ºC, washed, further incubated with 50 µl of

HRP-conjugated streptavidin at room temperature, washed, and developed with 50 µl of TMB-ELISA substrate. The reaction was stopped by the addition of 2 N H$_2$SO$_4$, and the resulting color developed was measured at 450 nm in a Synergy H1 plate reader (Bio-Tek).

For Lev-Lev5 binding assay, levan, inulin and glucan (Sigma) were immobilized in the wells of a MaxiSorp 96-well flat-bottom Nunc plate (2 µg per well) overnight at 37 °C. Subsequently, the wells were washed twice with 200 µl TBST and once with TBS, blocked with 3% BSA for 4 h at room temperature, and washed. For binding assays, two-fold dilutions of Lev-Lev5 starting from 0.4 mg/mL were prepared and added to the wells (50 µl per well) in duplicate. Buffer was added instead of Lev-Lev5 in control wells. After 2 h of incubation at room temperature, wells were washed (thrice with TBST and twice with TBS), incubated with mouse anti-C-terminal-His antibody (R930-25, Invitrogen) (50 µl per well) for an hour followed by washing and incubation with HRP-conjugated donkey anti-mouse IgG (715-035-150, Jackson Immunoresearch) (50 µl per well) for an hour at room temperature. TMB-ELISA substrate was added (50 µl per well), the reaction was stopped by adding 2 N H2SO4 (50 µl per well), and the resulting color developed in each well was measured at 450 nm in a Synergy H1 plate reader (Biotek). Non-linear curve fitting of the data was performed using Sigma plot software.

For Lev-Lev5 competitive inhibition assay, eleven polysaccharides (levan, inulin, xylan, amylopectin, pectin, arabinogalactan, glucan, dextran, laminarin, amylose and starch; 10 µg/mL concentration) were coated on MaxiSorp flat-bottom 96 well Nunc plate overnight at 37 °C. Buffer was added instead of polysaccharides in control wells. Subsequently, the wells were washed twice with 200 µl TBST and once with TBS, blocked with 3% BSA for 4 h at room temperature, and washed. Lev-Lev5 (5 µg/mL) was pre-incubated for an hour with buffer or with 10 mM sugars (corresponding to the constituent monosaccharide sugars of the polysaccharides coated in the well), and then added into the wells with coated polysaccharides and incubated at room temperature for 2 h. Wells were washed (thrice with TBST and twice with TBS), incubated with mouse anti-C-terminal-His antibody (R930-25, Invitrogen) (50 µl per well) for an hour followed by washing and incubation with HRP-conjugated donkey anti-mouse IgG (715-035-150, Jackson Immunoresearch) (50 µl per well) for an hour at room temperature. TMB-ELISA substrate was added (50 µl per well), the reaction was stopped by adding 2 N H2SO4 (50 µl per well), and the resulting color developed in each well was measured at 450 nm in a Synergy H1 plate reader (Biotek).

For St-Glc1 and St-Glc5v2 binding assay with starch, amylose and amylopectin, the wells of a Costar 3598 96-well flat bottom plate were coated with 1 µg of starch, amylose or amylopectin (by the addition of 100 µl of aqueous 10 µg/ml stocks made from 1 mg/ml starch in DMSO, 1 mg/ml amylose in DMSO, and 1 mg/ml amylopectin in water), and dried overnight at 37 °C. Subsequently, the wells were washed twice with 200 µl TBST and once with TBS, blocked with 3% BSA for 4 h at room temperature, and washed. For binding assays, two-fold dilutions of St-Glc1 (starting from 0.3 mg/ml) and St-Glc5v2 (starting from 0.56 mg/ml) were added to the wells. After 2 h of incubation at room temperature, wells were washed (thrice with TBST and twice with TBS), incubated with mouse anti-polyHistidine antibody (H1029, Sigma) (50 µl per well) for an hour followed by washing and incubation with HRP-conjugated donkey anti-mouse IgG (715-035-150, Jackson Immunoresearch) (50 µl per well) for an hour at room temperature, followed by washing. TMB-ELISA substrate was added (50 µl per well), the reaction was stopped by adding 2 N H2SO4 (50 µl per well), and the resulting color developed in each well was measured at 450 nm in a Synergy H1 plate reader (Biotek). Non-linear curve fitting of the data was performed using Sigma plot software.

For St-Glc1 and St-Glc5v2 competitive inhibition assays with starch, amylose, and amylopectin, the wells of a Costar 3598 96-well flat bottom plate were coated with 1 µg of starch, amylose or amylopectin and blocked with 3% BSA as described above. Pre-incubation of St-Glc1 (0.06 mg/ml) and St-Glc5v2 (0.11 mg/ml) with 100 µg/ml starch, amylose, amylopectin and glucose was performed at 4 °C for 30 min. Fivefold dilutions of the proteins (with or without prior incubation with starch, amylose, or amylopectin) were added to the wells in triplicate and incubated for 2 h at room temperature. Wells were then washed, and bound protein detected by incubation with mouse anti-polyHistidine antibody (H1029, Sigma) followed by HRP-conjugated donkey anti-mouse IgG (715-035-150, Jackson Immunoresearch) as described above.

**Statistics and reproducibility.** Biochemical assays were replicated, means (and standard deviations where relevant) are plotted and details of number of replicates are mentioned in the figure legends. Screening with phage display library was performed only once because our goal was to find the carbohydrate binding sequences, and later confirm the binding by performing biochemical assays.

**Reporting summary.** Further information on research design is available in the Nature Portfolio Reporting Summary linked to this article.

## Data availability

Sequences identified in this study have been deposited in GenBank with the following accession numbers—MN313892, MN313893, MN313894, MN313895, ON711254, ON711255, ON711256, ON711257, ON711258, ON711259, ON711260, ON711261,

ON711262, ON711263, ON711264, ON711265, ON711266, ON711267, ON711268, ON711269, ON711270, ON711271, ON711272, ON711273, ON711274, ON711275, ON711276, ON711277, ON711278, and ON711279. The metagenomic phage display library and the plasmids described in the study are available upon request and payment of shipping charges; however, individual phage clones described in the study are no longer viable, and the diversity of the phage display library is likely to have decreased with the passage of time during storage.

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

## Acknowledgements

This work was supported by grants awarded by the Council of Scientific Research (CSIR) - BSC0119 (Twelfth Five Year Plan Network Project titled "Man as a super-organism: Understanding the Human Microbiome" awarded to TNCR and SS2) and OLP0178 (Research Council approved project titled "Exploring the function of novel microbial carbohydrate binding domains" awarded to TNCR). A.A. and M.L. acknowledge the Council of Scientific and Industrial Research (CSIR) for their fellowships, S. Sunsunwal and Kajal acknowledge the University Grants Commission for their fellowship, and A.Y. acknowledges the Department of Biotechnology (DBT), Government of India, for DBT-BINC-Junior Research fellowship. The funders had no role in study design, data collection and analysis, decision to publish, or preparation of the manuscript. The authors acknowledge Prof. F. William Studier, Brookhaven National Laboratory and Dr. Tatjana Heinrich, Institute for Child Health Research, Western Australia for the kind gifts of *E. coli* BL21 pAR3924,5453, *E. coli* BL24 pAR3924,5453, and the lysate of replication-deficient T7 Δ9-10B, D104, Δ38 phage, and the protocol for making T7 packaging extract, Mr. Zachary Klamer, Van Andel Institute, Michigan, for guidance in the use of MotifFinder for glycan array data analysis, the Protein-Glycan Interaction Resource of the CFG (supporting grant R24 GM098791) and the National Center for Functional Glycomics (NCFG) at Beth Israel Deaconess Medical Center, Harvard Medical School (supporting grant P41 GM103694) for the glycan array analysis, the CSIR-IMTECH mass spectrometry facility and the Taplin Mass Spectrometry Facility for mass spectrometry analysis, and CSIR-IMTECH (manuscript communication number 020/2022) for the research facilities and infrastructure.

## Author contributions

R.T.N.C. conceived the study. A.A. performed all experiments relating to metagenomic DNA isolation, phage display metagenomic library preparation, screening of the phage display library against all glycoconjugates, identification of enriched phage clones, cloning of MG1, MN3, MU1, and MU3 metagenomic inserts in pET-28a(+), and protein expression and purification of these recombinant proteins for glycan array analysis. M.L. performed biochemical characterization of MG1, MN3, MU1, and MU3, including enzyme linked lectin assays and all isothermal calorimetry experiments. S. Sunsunwal performed cloning into the expression vector for all metagenomic inserts other than MG1, MN3, MU1, and MU3, and protein expression and purification for all these clones other than St-Glc1 and St-Glc5/St-Glc5v2, and enzyme linked lectin assays for Lev-Lev5, StapPG-GlcNAc6, and PBGal-Gal8. A.Y. performed bioinformatics analysis of the metagenomic sequences. Kajal performed protein expression, purification, and enzyme linked lectin assays for St-Glc1 and St-Glc5v2. S. Subramanian performed the structure analysis of the metagenomic sequences. A.A. and R.T.N.C. wrote the first draft of the manuscript. All authors participated in experimental design, data analysis, and manuscript preparation and editing.

## Competing interests

The authors declare no competing interests.
