## [Peer Review File · Communications Biology]

Reviewers' comments:

Reviewer #1 (Remarks to the Author):

The manuscript provides efforts to discover previously uncharacterized carbohydrate binding enzymes from human gut metagenomic samples. Method-wise, the work is well executed and offers new insights into the interactions between the gut microbiome and carbohydrates. However, the structure and style of the work need attention. Large parts of the results are very repetitive and can be drastically summarized. Likewise I feel figures 3,4, 5, and 6 can be combined.

Other comments:

General:

- Please use row numbers during for reviewing.
 - Many sentences across the paper are very long and thereby hard to follow. For example: 'Considering that the median size of bacterial proteins is 267 amino acids that CBMs are typically about 30 to 200 amino acids long, that T7 phage display systems have less sequence bias as compared to M13 and can be used to display proteins of about 1200 amino acids, and that individual protein-carbohydrate interactions are typically weak and require multivalency for optimal binding strength, we chose to construct the phage display library using the commercially available phage vector T7Select 10-3b to maximize on the display size (~1200 amino acid long peptides) with 5 sufficient copy number of peptides (5-15 per phage) to enable avidity of binding and thereby efficient screening.'
- Which is over 100 words. Please separate such long sentences into smaller clauses to improve legibility.

Introduction:

- From 'These microbes...' – Do only uncultivable bacteria provide a mineable reservoir for finding novel enzymes? I'd say also proteins of unknown function in cultivable bacteria (of which there are many) offer this opportunity.
- 'It is evident...' – misspelled 'gut microbiota'

Results:

- 'We constructed a metagenomic...' – Can the authors give some more details on the subjects chosen (e.g. mean/median age, % who were female, geographical location). Further, perhaps add some detail on how the carbohydrates/glycoconjugates were selected.
- 'In order to...the eluting agent' – this whole section reads more like methods than results. Please shorten and/or move. The same is true for large parts of the 'Construction of human fecal metagenomic phage display library and biopanning against carbohydrates/glycoconjugate' section.
- 'by searching against Pfam 31, 32 or the dbCAN2 database 33, indicating the absence of any annotated CAzyme, CBM or other protein domain' – have the authors tried looking for distant homologs with HMMs using HMMsearch or HHpred?
- 'An all against all search' – what kind of search? Blastn, blastp?
- The section detailing the glycan-binding profiles of MG1, MN3, and MU3 feels quite repetitive, and would benefit from summarization into one paragraph.
- 'An all against all search indicated significant pairwise' – Here too, please detail whether homology was tested on DNA or AA level.

Discussion:

- From 'Here, we have performed functional...' to 'obtained in our levan binding screen' - this reads not as a discussion but a summary, please shorten considerably/remove.

Reviewer #2 (Remarks to the Author):

The manuscript describes the use of phage display for identifying carbohydrate-binding domains originating from organisms from the health gut. The benefit of using phage display for the

metagenomics screens is that proteins originating from organisms hard to culture. The approach not only rely on the phage display, selected proteins were expressed recombinantly and binding of the produced proteins were tested. This is a clear strength of the study, as this take away the concern of false positives (the authors have also done a nice pre-test of the phage display setup to address the problem of false positives).

I found the approach and results very interesting, though the manuscript was very heavy to read, despite the English being very good. There was a lot of repeated text (it seems like the authors used a template for each section – especially the sections where recombinant production was described), where a protein name and carbohydrate type was exchanged – but else the rest of the text was the same. Furthermore, there were many different ligands mentioned in the text and I was some time losing the overview of which ligands the proteins bind and which they did not bind. I think that the section on page 23 starting with “We found that MG1, MN3...”, gives the reader a very good overview of the results. Please consider to rewrite parts of the results section to follow this “style. Furthermore, I will highly recommend that the results from each group of carbohydrates are presented in tables. Hence, the reader can get a quick overview of ligands tested – and to which binding was seen. I like the figures (Figures 3-6), where there are small structures of the ligands – as this gives a quick overview.

I think that there is no need to show all the agarose gels and SDS-PAGEs in the main text – they can be moved to suppl. materials. This will give room for tables/figures, which can help the reader get a better overview of the results (as mentioned above).

Now that you have identified some binding proteins, which could represent new families, would it be possible to use the DNA sequence of the identified proteins to get the full-length protein from the DNA purified from the fecal samples (assuming that the binding domains are found N-terminally to the catalytic domain)? I guess that the binding proteins could originate from some (novel) interesting enzymes?

Minor comments and questions (I have not read the manuscript carefully with regard to language/typos):

Please define “CBM” the first time used.

Page 3 – second last line: Correct “guzt” to “gut”

Throughout the text a mix is used when writing greek letters: please use the greek letters (α and β) instead of “a” and “b” or “alpha” and “beta”.

It was not clear to me if any of the recombinantly produced proteins by chance included a catalytic domain together with the carbohydrate-binding domain (some of the proteins were relatively large) - if so, did you test activity?

Page 34 – “E. coli” is missing in front of “TOP 10” and “BL21(DE3)” (maybe also other places?)

Page 34: The section starting with “For expression of recombinant...” is not very easy to follow.

Page 35: Please exchange “OD280” with “absorbance at 280 nm”

Page 36: Was the size exclusion chromatography done as a part of the purification or just to determine molecular weight of the proteins? If it was done as a part of the purification, then please move the section to the section, where the protein purification is described.

Page 38: You define TBS and TBS-T several times – it is enough to do it the first time the buffer abbreviations is used. Actually they are already defined on page 33.

Point-by-point Responses to Referees' comments

We are very thankful to both the reviewers for the rigorous and thorough review of our manuscript and for the suggestions, which we feel have helped improve our manuscript. Our point-by-point responses to the reviewers' comments follow below.

Reviewer #1 (Gut microbiota and Phage) Remarks to the Author:

The manuscript provides efforts to discover previously uncharacterized carbohydrate binding enzymes from human gut metagenomic samples. Method-wise, the work is well executed and offers new insights into the interactions between the gut microbiome and carbohydrates. However, the structure and style of the work need attention. Large parts of the results are very repetitive and can be drastically summarized. Likewise I feel figures 3,4, 5, and 6 can be combined.

Response: Thank you for the suggestions. We have rewritten the entire results section to summarize and avoid repetition. We have also combined figures 3, 4, 5, and 6 into one figure (figure 3 in the revised manuscript).

Other comments:

General:

- Please use row numbers during for reviewing.

Response: We have inserted row numbers in the revised manuscript.

- Many sentences across the paper are very long and thereby hard to follow. For example: 'Considering that the median size of bacterial proteins is 267 amino acids that CBMs are typically about 30 to 200 amino acids long, that T7 phage display systems have less sequence bias as compared to M13 and can be used to display proteins of about 1200 amino acids, and that individual protein-carbohydrate interactions are typically weak and require multivalency for optimal binding strength, we chose to construct the phage display library using the commercially available phage vector T7Select 10-3b to maximize on the display size (~1200 amino acid long peptides) with 5 sufficient copy number of peptides (5-15 per phage) to enable avidity of binding and thereby efficient screening.' Which is over 100 words. Please separate such long sentences into smaller clauses to improve legibility.

Response: We have broken this sentence (line 90 in revised manuscript), and other long sentences into smaller ones.

Introduction:

- From 'These microbes...' – Do only uncultivable bacteria provide a mineable reservoir for finding novel enzymes? I'd say also proteins of unknown function in cultivable bacteria (of which there are many) offer this opportunity.

Response: Indeed, we agree that proteins of unknown function in cultivable bacteria also offer this opportunity. We only state that "uncultured microbes represent an enormous

untapped biological resource that is largely unknown and that serves as a potential reservoir for mining novel enzymes and biomolecules”. We do not state that they are the only reservoir for mining novel enzymes and biomolecules, and so we are not implying that other microbes do not offer such an opportunity. Further, proteins of cultivable bacteria would also be represented in the metagenome.

- 'It is evident...' – misspelled 'gut microbiota'

Response: We have corrected the spelling (line 69) in the revised manuscript.

Results:

- 'We constructed a metagenomic...' – Can the authors give some more details on the subjects chosen (e.g. mean/median age, % who were female, geographical location). Further, perhaps add some detail on how the carbohydrates/glycoconjugates were selected.

Response: Details on the subjects chosen are provided in the Methods section “Sample collection” of the revised manuscript. A brief rationale on selection of subjects and selection of carbohydrates/glycoconjugates is provided in the Results section of the revised manuscript (lines 99-111).

- 'In order to...the eluting agent' – this whole section reads more like methods than results. Please shorten and/or move. The same is true for large parts of the 'Construction of human fecal metagenomic phage display library and biopanning against carbohydrates/glycoconjugate' section.

Response: Thank you for the suggestion. These sections have been shortened in the results section (lines 112-158) and relevant details moved to the methods section of the revised manuscript.

- 'by searching against Pfam 31, 32 or the dbCAN2 database 33, indicating the absence of any annotated CAZyme, CBM or other protein domain' – have the authors tried looking for distant homologs with HMMs using HMMsearch or HHpred?

Response: We agree that we did not explicitly look for distant homologs for the protein sequences using the above tools. However, for structure prediction, the tool we employed - Alphafold2 - implicitly performs sequence searches against protein databases using remote homology tools - hhsearch and hhblits - as a part of its MSA building step. And, based on the above predicted structures, we are able to make associations with existing glycan related protein folds. Besides, the sequence search method we employed - mmseqs2 - was run at a very high setting of sensitivity and the use of remote homology method is expected to increase the chances of returning false positives.

- 'An all against all search' – what kind of search? Blastn, blastp?

Response: It was an all against all protein sequence search. We have added this detail in the revised manuscript (lines 182, 241, 282, 320, 356).

- *The section detailing the glycan-binding profiles of MG1, MN3, and MU3 feels quite repetitive, and would benefit from summarization into one paragraph.*

Response: Thank you for the suggestion. This section has been re-written by summarizing the results of MG1, MN3, MU1, and MU3 together (lines 193-231).

- *'An all against all search indicated significant pairwise' – Here too, please detail whether homology was tested on DNA or AA level.*

Response: We have added this detail in the revised manuscript (lines 182, 241, 282, 320, 356).

Discussion:

- *From 'Here, we have performed functional...' to 'obtained in our levan binding screen' - this reads not as a discussion but a summary, please shorten considerably/remove.*

Response: This section has been shortened in the revised manuscript (lines 409-418).

Reviewer #2 (Protein Chemistry and Enzymology) Remarks to the Author:

The manuscript describes the use of phage display for identifying carbohydrate-binding domains originating from organisms from the health gut. The benefit of using phage display for the metagenomics screens is that proteins originating from organisms hard to culture. The approach not only rely on the phage display, selected proteins were expressed recombinantly and binding of the produced proteins were tested. This is a clear strength of the study, as this take away the concern of false positives (the authors have also done a nice pre-test of the phage display setup to address the problem of false positives). I found the approach and results very interesting, though the manuscript was very heavy to read, despite the English being very good. There was a lot of repeated text (it seems like the authors used a template for each section – especially the sections where recombinant production was described), where a protein name and carbohydrate type was exchanged – but else the rest of the text was the same. Furthermore, there were many different ligands mentioned in the text and I was some time losing the overview of which ligands the proteins bind and which they did not bind. I think that the section on page 23 starting with "We found that MG1, MN3...", gives the reader a very good overview of the results. Please consider to rewrite parts of the results section to follow this "style.

Response: Thank you for the suggestions. We have rewritten the entire results section to follow this style, and summarize and avoid repetition.

Furthermore, I will highly recommend that the results from each group of carbohydrates are presented in tables. Hence, the reader can get a quick overview of ligands tested – and to which binding was seen. I like the figures (Figures 3-6), where there are small structures of the ligands – as this gives a quick overview. I think that there is no need to show all the agarose gels and SDS-PAGEs in the main text – they can be moved to suppl. materials. This

will give room for tables/figures, which can help the reader get a better overview of the results (as mentioned above).

Response: Thank you for the suggestion. We have combined Figures 3-6, retaining the ligand structures for a quick overview, and introduced new tables (Table 1 and Table 2 in the revised manuscript) to give a quick overview of ligands tested and to which binding was seen in the revised manuscript. We have also removed some of the figures with agarose gels to the supplementary data file, and re-organized to show more data in less space.

Now that you have identified some binding proteins, which could represent new families, would it be possible to use the DNA sequence of the identified proteins to get the full-length protein from the DNA purified from the fecal samples (assuming that the binding domains are found N-terminally to the catalytic domain)? I guess that the binding proteins could originate from some (novel) interesting enzymes?

Indeed, using the DNA sequences of the identified CBMs (which are likely N-terminal or C-terminal to a catalytic domain), it should be possible to get the entire sequence of the full-length protein. However, this would be a lot of work and comprise a separate study. We mention the possibility of such a future study in the discussion (line 500 in the revised manuscript). We would like to note that the clone, Xln-Xyl8, itself has a GH10 and a CBM4/CBM16, and a GH10 and a carbohydrate binding domain (CM4/CBM9/CBM16/CBM22) as per dbCAN2 and Pfam database searches, respectively.

Minor comments and questions (I have not read the manuscript carefully with regard to language/typos):

Please define “CBM” the first time used.

Thank you for pointing this out. We have defined “CBM” on first use in the revised manuscript (line 71).

Page 3 – second last line: Correct “guzt” to “gut”

Thank you. We have corrected the spelling in the revised manuscript (line 69).

Throughout the text a mix is used when writing greek letters: please use the greek letters (α and β) instead of “a” and “b” or “alpha” and “beta”.

We have used the Greek letters now throughout the revised manuscript.

It was not clear to me if any of the recombinantly produced proteins by chance included a catalytic domain together with the carbohydrate-binding domain (some of the proteins were relatively large) - if so, did you test activity?

The clone, Xln-Xyl8, has a GH10 and a CBM4/CBM16, and a GH10 and a carbohydrate binding domain (CM4/CBM9/CBM16/CBM22) as per dbCAN2 and Pfam database searches, respectively. However, we did not test enzyme activity.

Page 34 – “E. coli” is missing in front of “TOP 10” and “BL21(DE3)” (maybe also other places?)

We have corrected this oversight throughout the revised manuscript.

Page 34: The section starting with “For expression of recombinant...” is not very easy to follow.

We have re-written this section (line 734-782), breaking it down into sub-sections, in the revised manuscript.

Page 35: Please exchange “OD280” with “absorbance at 280 nm”

We have replaced OD280 with absorbance at 280 nm in the revised manuscript (line 773).

Page 36: Was the size exclusion chromatography done as a part of the purification or just to determine molecular weight of the proteins? If it was done as a part of the purification, then please move the section to the section, where the protein purification is described.

Yes, it was done as part of the purification, where sufficient purity was not obtained with just Ni-NTA metal ion affinity chromatography. This section has been moved to the section on protein purification in the revised manuscript (lines 766-774).

Page 38: You define TBS and TBS-T several times – it is enough to do it the first time the buffer abbreviations is used. Actually they are already defined on page 33.

We have defined it the first time, and used the abbreviation in the subsequent occurrences in the revised manuscript.

REVIEWERS' COMMENTS:

Reviewer #1 (Remarks to the Author):

The authors' edits to the manuscript have improved the readability of the study. My only comment on the rebuttal is on Lines 46-48:

I generally agree with your response to my earlier comment on this line (i.e., you are not saying only uncultured microbes are an untapped resource, and your metagenomics datasets likely also contain ORFs from cultured bacteria. In light of the latter fact, I think starting the introduction with a mention of non-cultured bacteria as a resource is misleading to the actual contents of the manuscript.

A further minor comment: on line 204, the '(' is not accompanied by a ')'

Reviewer #2 (Remarks to the Author):

I think that the authors have replied well to my comments. However, I still think that the results section of the manuscript would benefit from being shortened even more. One example is the section found in line 206 ("We found that MG1, MN3,...) to line 222 of the merged revised file. It still contains way too much information: Since the information can now be found in a table, I think that you should just tell the most important information (actually I very much like the section in the discussion, lines 428-439, where you address these data, as this is clear and simple).

For each result section, you should think of what should the take home message be; mention only the most important findings, and let the very interested reader check the full results in the suppl. material/tables.

In the discussion, I think that you should change the order of some of the text:

With regard to the section from line 450 to line 470: You write a long introduction text (lines 450-462), before you write something about your own results - I would recommend that you shift the order. This would also allow you to shorten the text, I believe.

With regard to Lines 471 to 480: Again you have a general introduction text, which I do not think are really an argument for your own suggestion (lines 477-480).

I think that the figures are fine, but the different panels of the figures are becoming really small. I need to zoom significantly in order to be able to read the text on most of the figures.

Point-by-point Responses to Referees' comments

We are very thankful to both the reviewers for the rigorous and thorough review of our manuscript and for the suggestions, which we feel have helped improve our manuscript. Our point-by-point responses to the reviewers' comments follow below.

REVIEWERS' COMMENTS:

Reviewer #1 (Remarks to the Author):

The authors' edits to the manuscript have improved the readability of the study. My only comment on the rebuttal is on Lines 46-48:

I generally agree with your response to my earlier comment on this line (i.e., you are not saying only uncultured microbes are an untapped resource, and your metagenomics datasets likely also contain ORFs from cultured bacteria. In light of the latter fact, I think starting the introduction with a mention of non-cultured bacteria as a resource is misleading to the actual contents of the manuscript.

We have revised the introduction to shift the focus from non-cultured bacteria.

A further minor comment: on line 204, the '(' is not accompanied by a ')'

Thank you. We have inserted the “)” in the revised manuscript.

Reviewer #2 (Remarks to the Author):

I think that the authors have replied well to my comments. However, I still think that the results section of the manuscript would benefit from being shortened even more. One example is the section found in line 206 ("We found that MG1, MN3,...) to line 222 of the merged revised file. It still contains way too much information: Since the information can now be found in a table, I think that you should just tell the most important information (actually I very much like the section in the discussion, lines 428-439, where you address these data, as this is clear and simple).

We note that the reviewer is of the opinion that the manuscript would benefit from being shortened even more. In the example of the text mentioned by the reviewer, all details of the ligand motifs bound and not bound by MG1, MN3, MU1 and MU3 are not readily

apparent from table 1, and the reader will not be able to easily discern these details from the figures. Therefore, we have not shortened this section. However, we have shortened the remaining results sections, wherever possible.

For each result section, you should think of what should the take home message be; mention only the most important findings, and let the very interested reader check the full results in the suppl. material/tables.

Thank you for the suggestion. We have ensured that this is so.

In the discussion, I think that you should change the order of some of the text:

With regard to the section from line 450 to line 470: You write a long introduction text (lines 450-462), before you write something about your own results - I would recommend that you shift the order. This would also allow you to shorten the text, I believe.

We have revised and shortened this as recommended.

With regard to Lines 471 to 480: Again you have a general introduction text, which I do not think are really an argument for your own suggestion (lines 477-480).

We have revised and shortened this as recommended.

I think that the figures are fine, but the different panels of the figures are becoming really small. I need to zoom significantly in order to be able to read the text on most of the figures.

We are providing high-resolution figures to enable zooming on the text in the figures.